

# Dynamically controlled ozone decline in the tropical mid-stratosphere observed by SCIAMACHY

Evgenia Galytska[1,2], Alexey Rozanov[1], Martyn P. Chipperfield[3,4], Sandip. S. Dhomse[3], Mark Weber[1], Carlo Arosio[1], Wuhu Feng[3,5], and John P. Burrows[1]

[1]Institute of Environmental Physics, University of Bremen, Bremen, Germany
[2]Department of Meteorology and Climatology, Taras Shevchenko National University of Kyiv, Kyiv, Ukraine
[3]School of Earth and Environment, University of Leeds, Leeds, UK
[4]National Centre for Earth Observation, University of Leeds, Leeds, UK
[5]National Centre for Atmospheric Science, University of Leeds, Leeds, UK

**Correspondence:** E. Galytska (egalytska@iup.physik.uni-bremen.de)

**Abstract.** Despite the recently reported beginning of a recovery in global stratospheric ozone ($O_3$), an unexpected $O_3$ decline in the tropical mid-stratosphere (around 30-35 km altitude) was observed in satellite measurements during the first decade of the 21st century. We use SCanning Imaging Absorption SpectroMeter for Atmospheric CHartographY (SCIAMACHY) measurements for the period 2004-2012 to confirm the significant $O_3$ decline. The SCIAMACHY observations also show that
the decrease in $O_3$ is accompanied by an increase in $NO_2$.

To reveal the causes of these observed $O_3$ and $NO_2$ changes, we performed simulations with the TOMCAT 3D Chemistry-Transport Model (CTM) using different chemical and dynamical forcings. For the 2004-2012 time period, the TOMCAT simulations reproduce the SCIAMACHY-observed $O_3$ decrease and $NO_2$ increase in the tropical mid-stratosphere. The simulations suggest that the positive changes in $NO_2$ (around 7% per decade) are due to similar positive changes in reactive odd nitrogen
($NO_y$), which are a result of a longer residence time of the source gas $N_2O$ and increased production via $N_2O + O(^1D)$. The model simulations show a negative change of 10% per decade in $N_2O$ that is most likely due to variations in the deep branch of the Brewer-Dobson Circulation (BDC). Interestingly, modelled annual mean 'age-of-air' (AoA) does not show any significant changes in the transport in the tropical mid-stratosphere during 2004-2012.

However, further analysis of model results demonstrate significant seasonal variations. During the autumn months (September-
October) there are positive AoA changes, that imply transport slowdown and a longer residence time of $N_2O$ allowing larger conversion to $NO_y$ which enhances $O_3$ loss. During winter months (January-February) there are negative AoA changes, indicating faster $N_2O$ transport and less $NO_y$ production. Although the changes in AoA cancel out when averaging over the year, non-linearities in the chemistry-transport interactions mean that the net negative $N_2O$ change remains.

## 1    Introduction

Stratospheric ozone ($O_3$) is one of the most important components of the atmosphere. It absorbs ultraviolet solar radiation, which is harmful to plants, animals and humans, and thereby plays a key role in determining the thermal structure and dynamics



of the stratosphere (Jacobson, 2002; Seinfeld and Pandis, 2006). The amount of $O_3$ in the stratosphere is controlled by a balance between photochemical production and loss mechanisms. However the atmospheric dynamics play an important role in determining the conditions at which these photochemical and chemical reactions take. As a result $O_3$ global distribution and inter-annual variability are governed by transport processes, e.g. the Brewer-Dobson Circulation (BDC). To set the scene

for our understanding of chemical $O_3$ variations in the tropical mid-stratosphere, we briefly discuss the mechanism of $O_3$ production and loss via catalytic $NO_x$ ($NO_x$=NO + $NO_2$) cycle and the role of nitrous oxide ($N_2O$).

Stratospheric $O_3$ is essentially formed in the regions where solar ultraviolet electromagnetic radiation is present (Chapman, 1930). The first mechanism proposed to explain its formation and loss is known as the Chapman cycle. $O_3$ is formed via photodissociation of molecular oxygen ($O_2$) mostly within the so-called Herzberg continuum (200-242 nm; Nicolet, 1981).

Absorption by $O_2$ at shorter wavelengths (e.g. Schumann-Runge bands, 175-200 nm) occurs at higher altitudes, i.e. in the upper stratosphere, mesosphere. In the mid-stratosphere the ultraviolet sunlight breaks apart an $O_2$ molecule to produce two oxygen (O) atoms:

$$O_2 + h\nu \xrightarrow{\lambda < 242\,nm} O + O \tag{R1}$$

Then each O atom combines with $O_2$ to produce $O_3$:

$$O + O_2 + M \rightarrow O_3 + M \tag{R2}$$

where M represents a third body. Reactions (R1) and (R2) occur continually whenever shortwave ultraviolet radiation is present in the stratosphere. As a consequence, the strongest $O_3$ production takes place in the tropical mid-stratosphere. Then $O_3$ is photolyzed with lower-energy photons in the Hartley bands (242-310 nm) to produce excited singlet oxygen ($O(^1D)$) or in the Huggins bands (310-400 nm) to produce ground-state atomic oxygen $O(^3P)$:

$$O_3 + h\nu \xrightarrow{242 > \lambda > 310\,nm} O_2 + O(^1D) \tag{R3a}$$

$$\xrightarrow{310 > \lambda > 400\,nm} O_2 + O(^3P) \tag{R3b}$$

An important aspect of $O_3$ photochemistry is that it is the major source of $O(^1D)$ in the stratosphere (R3a). $O(^1D)$ is rapidly quenched to the electronic ground state by collision with any third-body molecule, most likely $N_2$ or $O_2$ ($O(^1D) + M \rightarrow O + M$; Jacob, 1999). The final Chapman cycle reaction of $O_3$ and atomic oxygen (R4) is relatively slow and does not cause significant

$O_3$ loss, since $O_2$ recombines with O atom to regenerate $O_3$ via Reaction (R2).

$$O_3 + O \rightarrow O_2 + O_2 \tag{R4}$$

Importantly, other chemical cycles can catalyse this reaction.



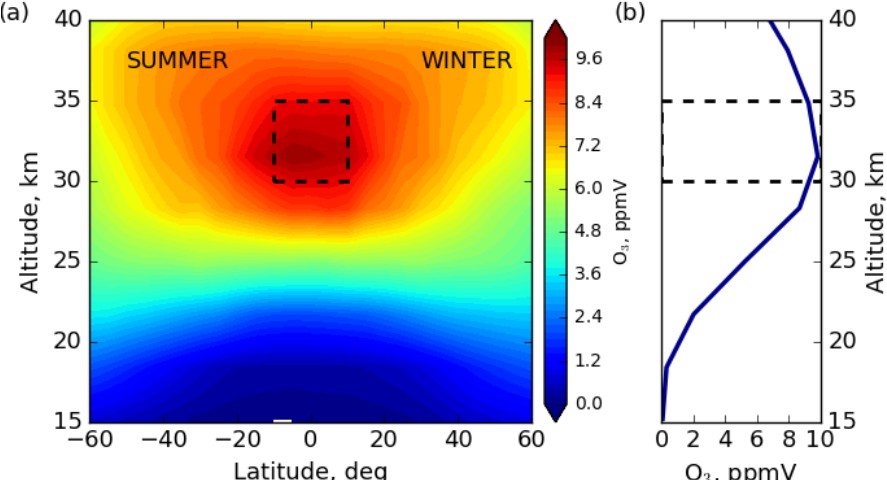

**Figure 1.** SCIAMACHY zonally averaged climatological mean $O_3$ (ppmV) for (a) latitude-altitude distribution, (b) profile during DJFs 2004–2012. The dashed rectangles indicate the region of tropical mid-stratosphere investigated in this paper.

The distribution of $O_3$ in the stratosphere is illustrated in Fig. 1. Panel a shows zonally-averaged climatological mean $O_3$ volume mixing ratio (vmr) as a function of latitude in the stratosphere from SCIAMACHY measurements (further described in Sect. 2.1) during boreal winters (December-January-February, DJF) 2004-2012. We chose Northern Hemisphere (NH) winter months to clearly represent the hemispheric distributions of $O_3$. Fig. 1b shows the mean $O_3$ vertical profile for the tropical region, averaged for the same period as in panel a. The dashed rectangles indicate the region of the tropical ($10°S$-$10°N$) mid-stratosphere (30-35 km) investigated in this study.

Significant $O_3$ destruction occurs through reaction with oxides of nitrogen ($NO_x$, whose major source is $N_2O$), hydrogen ($HO_x$=OH, $H_2O$, whose major sources are $CH_4$ and $H_2O$), chlorine ($ClO_x$, whose major sources are chlorofluorocarbons, known as CFCs, and other halocarbons) and bromine ($BrO_x$, whose major sources are methyl bromide and halons). Portmann et al. (2012) showed that the relative mean global $O_3$ loss in the upper and lower stratosphere is dominated by the $HO_x$, $ClO_x$/$BrO_x$ chemistry, and in the middle stratosphere by the $NO_x$ cycle, which is the largest near the $O_3$ maximum. Consequently, significant $O_3$ loss in the tropical mid-stratosphere is predominantly determined by catalytic $NO_x$ destruction (Crutzen, 1970), where NO rapidly reacts with $O_3$ to produce $NO_2$:

$$NO + O_3 \rightarrow NO_2 + O_2 \tag{R5}$$

$NO_2$ molecules can then react with (ground state) oxygen atoms:

$$NO_2 + O \rightarrow NO + O_2 \tag{R6}$$

In the middle stratosphere, the exchange time between NO and $NO_2$ in Reactions (R5) and (R6) is approximately one minute during daytime.





The primary source of $NO_x$ in the stratosphere is $N_2O$ (McElroy and McConnell, 1971), which is emitted at the surface by anthropogenic and microbial processes in the ocean and soils (Bregmann et al., 2000). $N_2O$ is an important greenhouse gas, inert in the troposphere, and is transported into the tropical stratosphere via the upwelling branch of the BDC. Around 90% of all $N_2O$ is photolyzed in the stratosphere by UV radiation between 180-230 nm (McLinden et al., 2003) with the maximum

absorption being in the region between 180-190 nm (Keller-Rudek et al., 2013):

$$N_2O + h\nu \xrightarrow{\lambda < 230\,nm} N_2 + O(^1D) \tag{R7}$$

The remaining 10% of $N_2O$ is removed by reaction with $O(^1D)$ which occurs through two channels. One of these (about 5% of overall $N_2O$ loss) contributes to NO production:

$$N_2O + O(^1D) \rightarrow NO + NO \tag{R8a}$$

$$\rightarrow N_2 + O_2 \tag{R8b}$$

We emphasise here that the oxidation of $N_2O$ via Reaction (R8a) is the primary source of NO (and $NO_x$) in the stratosphere, which then actively participates in $O_3$ destruction via (R5) and (R6).

The impact of $N_2O$ on both climate change and stratospheric $O_3$ is such that it is necessary to further control its emissions.

However, $N_2O$ is not included for regulation in the Montreal Protocol (WMO, 2014), signed in 1985 by the United Nations Vienna Convention for the Protection of the Ozone Layer to limit the negative impact of man-made $O_3$-depleting substances. Anthropogenic $N_2O$ is only regulated by the Kyoto Protocol to the United Nations Framework Convention on Climate Change (UNFCCC) and is expected to be the dominant contributor to $O_3$ depletion in the 21st century (Ravishankara et al., 2009).

Due to the long global lifetime of $N_2O$, which exceeds 100 years (e.g. Olsen et al., 2001; Seinfeld and Pandis, 2006;

Portmann et al., 2012; Chipperfield et al., 2014), its distribution is affected by changes in BDC. For example, Plummer et al. (2010) and Kracher et al. (2016) showed that the accelerated upwelling decreases the residence time of $N_2O$ in the stratosphere causing lower $NO_x$ production and as a consequence lower $O_3$ loss. For conditions of slower upwelling, the residence time of $N_2O$ in the stratosphere is expected to be longer, which then causes higher $NO_x$ production and higher $O_3$ loss.

Several publications in recent years have documented significant $O_3$ changes in the tropical mid-stratosphere, in particular

its decrease during the first decade of the 2000s (Kyrölä et al., 2013; Gebhardt et al., 2014; Eckert et al., 2014; Nedoluha et al., 2015b). Kyrölä et al. (2013) showed a statistically significant negative trend of $O_3$ of around 2-4% per decade for the period 1997-2011 at altitudes 30-35 km from the combined Stratospheric Aerosol and Gas Experiment (SAGE) II-Global Ozone Monitoring by Occultation of Stars (GOMOS) dataset. Gebhardt et al. (2014) identified negative $O_3$ trends of up to 10% per decade for the period August 2002-April 2012 at altitudes 30-38 km from SCanning Imaging Absorption spectroMeter for

Atmospheric CHartographY (SCIAMACHY) which are almost double the trends reported by Kyrölä et al. (2013). In addition, Gebhardt et al. (2014) pointed out a possible connection of negative $O_3$ trends with positive $NO_x$ changes (first presented at Quadrennial Ozone Symposium 2012). Eckert et al. (2014) reported negative $O_3$ trends in the tropics in a form of a doubled-peak structure at around 25 and 35 km from Michelson Interferometer for Passive Atmospheric Sounding (MIPAS) for the





period 2002-2012. However, the reasons for observed trends remained unclear, although Eckert et al. (2014) mentioned that the changes in upwelling explain neither the observed negative $O_3$ trends nor their doubled-peak structure.

The findings of Nedoluha et al. (2015b) were the most relevant to describe observed trends in the tropical mid-stratosphere. They showed a significant $O_3$ decrease at 30-35 km altitude in the tropics by using Halogen Occultation Experiment (HALOE; 1991-2005) and NASA Aura Microwave Limb Sounder (MLS; 2004-2013) data. They linked $O_3$ decrease with the long-term increase of the bulk of $NO_y$ ($NO_x$ + $HNO_3$ + $2\times N_2O_5$) species, which in turn they explained by changes in $N_2O$ transported from the troposphere. In particular they showed that the decrease in $N_2O$ is 'likely linked to long-term variations in dynamics'. Using a 2D chemical-dynamical model they showed that the simulated increase of tropical upwelling led to lower $N_2O$ oxidation via (R8a). As a consequence less $NO_y$ was produced which resulted in less $O_3$ destruction. With such conclusions Nedoluha et al. (2015b) argued that simulated dynamical perturbations could explain changes of $O_3$ in the tropical mid-stratosphere. Nevertheless, the authors did not show that such dynamical perturbations in the BDC indeed existed in the atmosphere. The changes in the strength of different BDC branches were analysed by Aschmann et al. (2014). They used diabatic heating calculations from the European Centre for Medium-Range Weather Forecasts (ECMWF) Era-Interim data set. They concluded that 'there are strong indications that the observed trend-change in $O_3$ is primarily a consequence of a simultaneous trend-change in tropical upwelling'. The conclusions of both Aschmann et al. (2014) and Nedoluha et al. (2015b) agree with the finding of Shepherd (2007), who showed that stratospheric $O_3$ is affected by variations in transport patterns, which in turn are associated with changes in Rossby-wave forcing.

The most recent publications with extended data records suggest that there are signs of $O_3$ recovery in tropical mid-stratosphere. Sofieva et al. (2017) showed small negative (2% per decade) $O_3$ trend by analysing merged SAGE II, European Space Agency (ESA) Ozone Climate Change Initiative (Ozone_cci) and Ozone Mapping Profiler Suite (OMPS) datasets for the period 1997-2016. Steinbrecht et al. (2017) analysed seven merged data sets and concluded that there are no clear indications for $O_3$ changes during 2000-2016. Ball et al. (2018) highlighted that the observed decrease of $O_3$ at 32-36 km is primarily due to high $O_3$ during 2000-2003 period and they did not report negative $O_3$ changes during 1986-2016. Positive $O_3$ trends in the tropical stratosphere above 10 hPa were shown in the most recent research of Chipperfield et al. (2018, Fig. 3) for the period 2004-2017 from MLS measurements and simulations of the TOMCAT CTM. While there are clear signs of recent recovery of stratospheric ozone layer (Chipperfield et al., 2017), full explanations of observed negative $O_3$ changes in the tropical mid-stratosphere within the first decade of the 21st century have not been quantified.

In this study we analyse changes in the tropical mid-stratosphere based on updated SCIAMACHY $O_3$ and $NO_2$ datasets during 2004-2012, which is similar to the period analysed by several studies (Kyrölä et al., 2013; Gebhardt et al., 2014; Eckert et al., 2014; Nedoluha et al., 2015b). However, in contrast to those studies, we combine and compare SCIAMACHY measurements with simulations of TOMCAT, a state-of-the-art 3D chemistry-transport model (CTM). We additionally perform TOMCAT runs with different chemical and dynamical forcings to diagnose the primary causes of $O_3$ and $NO_2$ changes. We also consider modelled $NO_x$, the major component of mid-stratosphere $NO_y$, and $N_2O$ species in our analysis. Based on modelled age-of-air (AoA) data we demonstrate seasonal changes in the deep branch of BDC. We further explain how transport changes from month-to-month affect $N_2O$ chemistry, which consequently leads to observed $O_3$ changes. Note that in this paper, we do



not refer to observed changes of chemical compounds in the mid-stratosphere as 'trends', as the analysed time span is not long enough. Consequently, we use the term 'changes' instead.

## 2 Methods and data sources

### 2.1 SCIAMACHY limb data

The ESA Environmental Satellite (Envisat) mission carried ten sensors dedicated to Earth observation, which were operational from the launch of the satellite in March 2002 until it failed in April 2012, doubling the planned lifetime of 5 years. Envisat was in a near-circular sun-synchronous orbit at an altitude of around 800 km, with the inclination of 98°. The SCIAMACHY instrument (Burrows et al., 1995; Bovensmann et al., 1999) onboard Envisat was a passive imaging spectrometer that comprised eight spectral channels and covered a broad spectral range from 240 to 2400 nm.

SCIAMACHY performed spectroscopic observations of solar radiation scattered by and transmitted through the atmosphere, as well as reflected by the Earth's surface in three viewing modes: limb, nadir, and occultation. We use only SCIAMACHY data in limb-viewing geometry in our study. In this case, the line of sight of the instrument follows a slant path tangentially through the atmosphere and solar radiation is detected when it is scattered into the field of view of the instrument. The limb geometry combines near-global coverage with a moderately high vertical resolution of about 3 km. SCIAMACHY scanned the

Earth's limb within a tangent height range of about -3 to 92 km (0 to 92 km since October 2010) in steps of about 3.3 km. The vertical instantaneous field of view of the SCIAMACHY instrument was ∼2.6 km, and the horizontal cross-track instantaneous field of view was ∼100 km at the tangent point. However, the horizontal cross-track resolution is mainly determined by the integration time during the horizontal scan resulting typically in a value of about 240 km. Global coverage of SCIAMACHY limb measurements was obtained within 6 days at the equator and less elsewhere.

$O_3$ and $NO_2$ profiles data used here are from IUP (Institut für Umweltphysik) Bremen limb retrievals Versions 3.5 and 3.1, respectively. Monthly mean $O_3$ (Jia et al., 2015) and $NO_2$ (Butz et al., 2006) data were gridded horizontally into 5°latitude × 15°longitude and vertically into ∼3.3 km altitude bins, covering the altitude range from 8.6 to 64.2 km.

We use both $O_3$ and $NO_2$ data for altitudes 15-40 km. The zonal monthly mean $O_3$ and $NO_2$ values were calculated as arithmetic means as the errors of single measurements are mostly normally distributed and no additional issues with outliers

have been reported. Zonal monthly mean values were typically composed of hundreds of single measurements. Consequently, we assumed that the random errors of zonal monthly means could be neglected. We chose the boundaries 60°S-60°N to circumvent gaps in SCIAMACHY sampling during polar winters.

### 2.2 TOMCAT model

We have performed a series of the experiments with the global TOMCAT offline 3-D CTM (Chipperfield, 2006). The model

contains a detailed description of stratospheric chemistry including species in the $O_x$, $HO_x$, $NO_y$, $Cl_y$ and $Br_y$ chemical families. The model includes heterogeneous reactions on sulfate aerosols and polar stratospheric clouds. The model was forced us-





ing ECMWF ERA-Interim winds and temperatures (Dee et al., 2011) and simulations were performed at $2.8° \times 2.8°$ horizontal resolution with 32 $\sigma$-p levels ranging from the surface to about 60 km. The surface mixing ratios of long-lived source gases (e.g. CFCs, HCFCS, $CH_4$, $N_2O$) were taken from WMO (2014) scenario A1. The solar cycle was included using time-varying solar flux data (1950-2016, Dhomse et al., 2016) from the Naval Research Laboratory (NRL) solar variability model, referred to

as NRLSSI2 (Coddington et al., 2016). Stratospheric sulfate aerosol surface density (SAD) data for 1850-2014 were obtained from ftp://iacftp.ethz.ch/pub_read/luo/CMIP6/ (Arfeuille et al., 2013; Dhomse et al., 2015).

We performed a total of three model simulations constrained for SCIAMACHY measurements to help distinguish the dynamical and chemical effects on stratospheric $O_3$ and $NO_2$. The control run (CNTL) was spun up from 1977 and integrated until the end of 2012 including all of the processes described above. Sensitivity simulations were initialised from the control

run in 2004 and also integrated until the end of 2012. Run fSG was the same as run CNTL but used constant tropospheric mixing ratios of all source gases after 2004. This removes the long-term trends in composition due to source gases changes. Run fDYN was the same as CNTL but used annually repeating meteorology from 2004. All of the simulations included an idealised stratospheric AoA tracer which was forced using a linearly increasing tropospheric boundary condition.

## 2.3  Multiple linear regression

To assess the temporal evolution of chemical compounds we applied a multiple linear regression (MLR) model similar to Gebhardt et al. (2014) to SCIAMACHY $O_3$ and $NO_2$ and TOMCAT $O_3$, $NO_2$, $N_2O$, $NO_x$, and $NO_y$ species time series for the period January 2004-April 2012. The MLR was performed for each latitude band and altitude level and included the following proxies: the seasonal variations (12- and 6-month terms), Quasi-Biennial Oscillation (QBO), El Niño–Southern Oscillation (ENSO), a constant, and linear terms as shown in Eq. (1):

$$\mu + \omega t + \sum_{j=1}^{2}(\beta_{1j}sin(\frac{2\pi jt}{12}) + \beta_{2j}cos(\frac{2\pi jt}{12})) + aQBO_{10}(t) + bQBO_{30}(t) + cENSO(t), \tag{1}$$

where $\mu$ stands for intercept of a linear fit of regression analysis, $\omega$ is its slope (linear changes). Time (in months) is represented by $t$, and varies from 1 to 100, where 1 corresponds to January 2004 and 100 to April 2012. $\beta_{11}...\beta_{22}$, $a$, $b$, and $c$ are additional fitting parameters. The harmonics with annual (12 months) and semi-annual (6 months) periods, which correspond j=1 and j=2, accordingly, are used to represent seasonal variations. The combination of sin and cos modulations adjusts to any

phase of the (semi-)annual variations. In the latitudes between 50-60°N and within altitude range 15-26 km we applied cumulative eddy heat flux instead of harmonic fit terms. We used ERA-Interim eddy heat flux at 50 hPa integrated from 45°N to 75 °N with the time lag of 2 months. $QBO_{10}(t)$ and $QBO_{30}(t)$ are the equatorial winds at 10 and 30 hPa, respectively (available from http://www.geo.fu-berlin.de/en/met/ag/strat/produkte/qbo/index.html). Monthly time series of equatorial winds were smoothed by a 4-month running average. $ENSO(t)$ - represents ENSO and is based on anomalies of the Nino 3.4 index (available from

http://www.cpc.ncep.noaa.gov/data/indices/). In our regression model ENSO is accounted within the latitude band 20°S-20°N and at altitudes 15-25 km with a time lag of 2 months. In addition to above-mentioned proxies, we have calculated changes both with and without the solar cycle term. The solar cycle term is represented by multi-instrument monthly mean Mg II index from





GOME, SCIAMACHY, and GOME-2 (available from http://www.iup.uni-bremen.de/gome/solar/MgII_composite.dat, Weber et al., 2013). The results with and without solar cycle term are very similar. Therefore, we only show results from MLR without a solar cycle term.

As the noise autocorrelation is not applicable when calculating linear changes for selected months, so for reasons of consistency it was also ignored for linear changes from the complete time series. We used the $1\sigma$ value, which is defined by a covariance matrix of regression coefficients, as the uncertainty of observed changes. The significance of observed changes at the 95% confidence level is met if the absolute ratio between the trend and its uncertainty is larger than 2 (Tiao et al., 1990). For all chemical species, we show changes in relative units with respect to the mean value of the whole time series, i.e. % per decade. Changes of AoA are shown in absolute values, i.e. years per decade.

## 3 Results and discussion

### 3.1 Observed and simulated changes from SCIAMACHY and TOMCAT

Figure 2 shows latitude-altitude plots of the $O_3$ and $NO_2$ linear changes from SCIAMACHY measurements over the latitude range 60°S-60°N and altitude range 15-40 km during January 2004-April 2012. Hatched areas show regions where changes are significant at the $2\sigma$ level. The plot is based on zonal monthly mean values with data gridding as described in Sect. 2.1. Statistically significant positive $O_3$ changes of around 6% per decade are observed at southern mid-latitudes at altitudes around 27-31 km (Fig. 2a), which agree well with linear $O_3$ trends from MLS for the period 2004-2013 shown by Nedoluha et al. (2015b). More pronounced positive $O_3$ changes are seen in the tropical lower stratosphere up to ~22 km altitude, which match well with results reported by Gebhardt et al. (2014) and Eckert et al. (2014). However, the focus of our analysis remains on the region of the tropical mid-stratosphere bounded by the dashed rectangle in Fig. 2. This is the region where the 'island' of statistically significant negative $O_3$ changes is observed, reaching around 10% per decade.

The SCIAMACHY version 3.5 $O_3$ data (see Sect. 2.1) used in this study employs an updated retrieval approach in the visible spectral range (Jia et al., 2015) compared to older data versions. The observed negative $O_3$ changes in the tropical mid-stratosphere (Fig. 2) are consistent with Gebhardt et al. (2014), who applied version 2.9 $O_3$ data during a similar period (August 2002-April 2012), using a similar regression model. Such negative $O_3$ changes also agree well with the findings of Kyrölä et al. (2013); Eckert et al. (2014); Nedoluha et al. (2015b); Sofieva et al. (2017), albeit they employed different datasets within similar, but not identical, time spans. Figure 2b shows a strong positive change in $NO_2$ of around 15% per decade in the region of the tropical mid-stratosphere.

To identify possible reasons for the $O_3$ changes in the tropical mid-stratosphere, and to check the role of $N_2O$ and $NO_x$ chemistry in these changes following suggestions by Nedoluha et al. (2015b), we analyse data from three TOMCAT simulations (see Sect. 2.2). Figure 3 presents latitude-altitude plots of the linear changes in $O_3$ (panels a-c), $NO_2$ (panels d-f), and $N_2O$ (panels g-i) for the period January 2004–April 2012 from TOMCAT (1) control run, CNTL - left column, (2) run with constant tropospheric mixing ratios of source gases, fSG - middle column, and (3) run with annually repeating meteorology, fDYN - right column. Results are shown on the native TOMCAT vertical grid. Latitude-altitude plots of equivalent $NO_x$ and $NO_y$



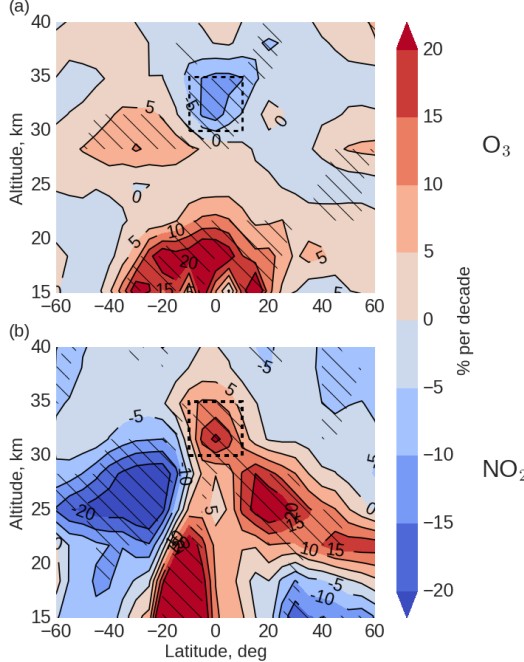

**Figure 2.** Latitude-altitude distribution of (a) $O_3$ and (b) $NO_2$ changes (% per decade) from MLR model of SCIAMACHY measurements for January 2004–April 2012. Hatched areas show changes significant at the $2\sigma$ level. The dashed rectangle indicates the region of the tropical mid-stratosphere investigated in this paper.

linear changes from TOMCAT are shown in the Supplements Fig. S1–S2. The CNTL simulation shows negative $O_3$ changes in the tropical mid-stratosphere (Fig. 3a) of around 5% per decade and positive $NO_2$ changes (Fig. 3d) of around 10% per decade, which are similar, but somewhat smaller than changes observed by SCIAMACHY (Fig. 2a,b). Figure 3g indicates statistically significant $N_2O$ decrease of around 15% per decade in the tropical mid-stratosphere and pronounced hemispheric asymmetry

5 with positive changes at southern and negative changes at northern mid-latitudes. These changes agree well with $N_2O$ trends from MLS during 2004-2013 (see Nedoluha et al., 2015a, Fig. 10). Such variations of $N_2O$, a long-lived tracer with the global lifetime of around 115-120 years (Portmann et al., 2012), might indicate possible changes in the deep branch of the BDC.

To distinguish the role of transport on $O_3$, $NO_2$, and $N_2O$ changes in the tropical mid-stratosphere, we show the results of the TOMCAT fSG simulation with the constant tropospheric mixing ratios of all source gases in the middle column of Fig.

10 3. The modelled changes from both runs CNTL and fSG are very similar for $O_3$ (Fig. 3a,b), $NO_2$ (Fig. 3d,e), and $N_2O$ (Fig. 3g,h). This illustrates that the observed changes in the tropical mid-stratosphere are mostly of dynamical origin. The TOMCAT fDYN simulation, with annually repeating meteorology, shows insignificant negative changes in $O_3$ (Fig. 3c). Both $NO_2$ (Fig. 3f) and $N_2O$ (Fig. 3i) show statistically significant but very weak positive changes in the tropical mid-stratosphere of around 1-3% per decade. This indicates that the direct impact of the chemistry on observed variations of $O_3$, $NO_2$, and $N_2O$ is small.





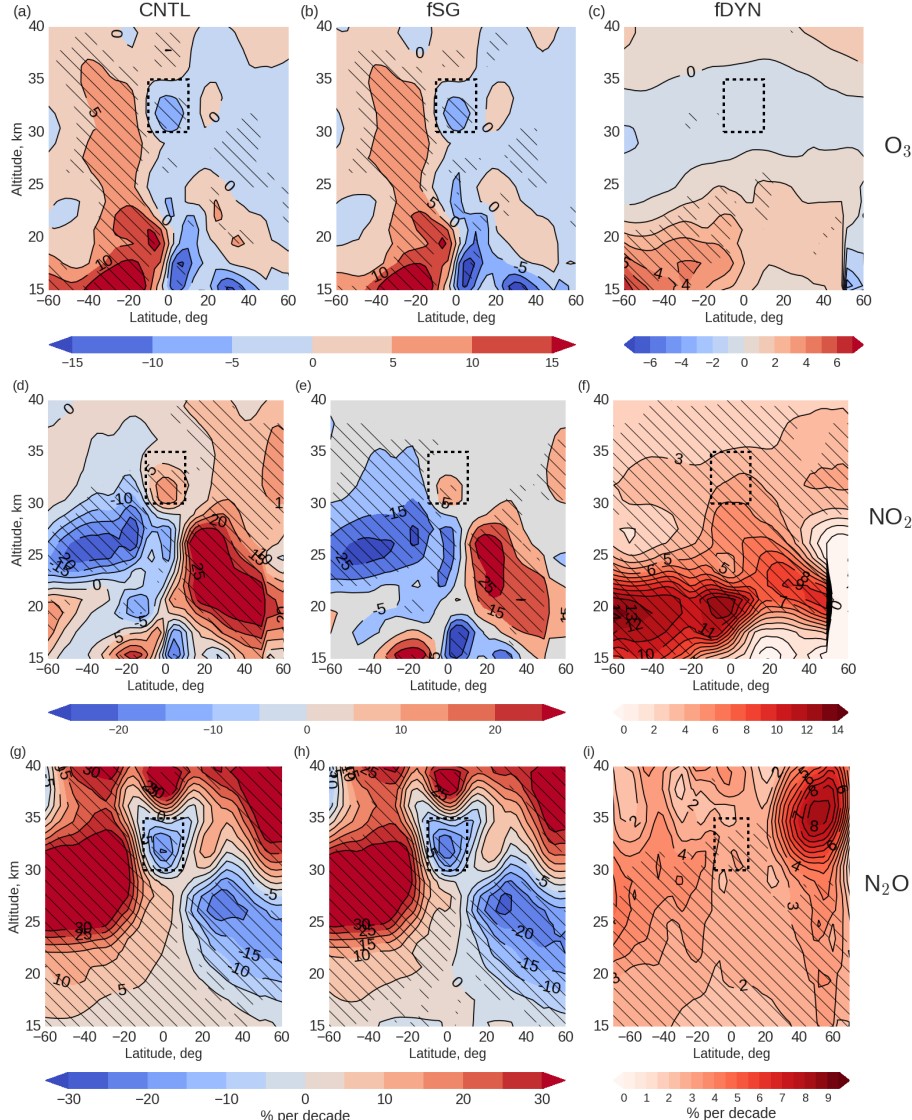

**Figure 3.** Latitude-altitude distribution of (a-c) $O_3$, (d-f) $NO_2$, and (g-i) $N_2O$ changes (% per decade) from the MLR model applied to TOMCAT runs for January 2004–April 2012 period: CNTL (left column), fSG (middle column), and fDYN (right column): latitude range from $60°$S to $60°$N, altitude range from 15 to 40 km. Hatched areas show changes significant at the $2\sigma$ level. The dashed rectangle indicates the region of the tropical mid-stratosphere investigated in this paper.

## 3.2 Tropical mid-stratospheric correlations

A powerful diagnostic for identifying the impact of chemical and dynamical processes on specific stratospheric constituents is tracer-tracer correlation plots (e.g. Sankey and Shepherd, 2003; Hegglin and Shepherd, 2007). Figure 4 shows correlation plots





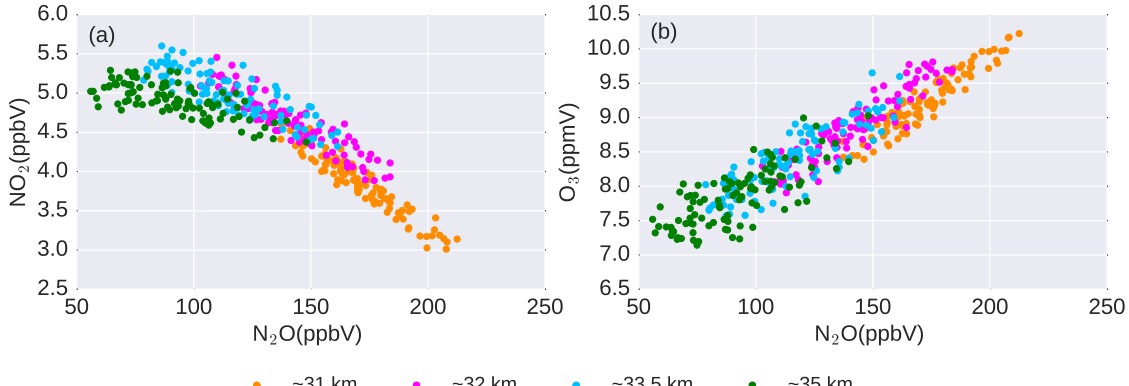

**Figure 4.** Scatter plots of monthly mean (a) $N_2O$ versus $NO_2$ and (b) $N_2O$ versus $O_3$ in the tropical mid-stratosphere during January 2004-April 2012 from TOMCAT simulation CNTL. Colour coding classifies data according to altitude: 31 km (orange), 32 km (magenta), 33.5 km (sky blue), and 35 km (green).

of $N_2O$ versus $NO_2$ and $N_2O$ versus $O_3$ in the tropical mid-stratosphere from the TOMCAT run CNTL. The colour coding classifies data according to altitude: 31 km (orange), 32 km (magenta), 33.5 km (sky blue), and 35 km (green). The panels show monthly mean data over the period January 2004-April 2012. Due to its long lifetime, $N_2O$ values reflect transport patterns: low $N_2O$ values indicate older air and high $N_2O$ values younger air. Figure 4a shows $N_2O$-$NO_2$ anti-correlation, which results

from $N_2O$ chemical loss to produce $NO_2$ and which is coupled with dynamical impact. Namely, when the upwelling is speeded up, more $N_2O$ is transported from lower altitudes, but less $NO_2$ (and therefore $NO_y$) is formed as the residence time of $N_2O$ decreases. Consequently, there is less time to produce $NO_2$ via Reaction (R8a). In contrast, with slower upwelling less $N_2O$ is transported to the mid-stratosphere, but its residence time is longer which allows increased $NO_2$ production. Fig. 4a, clearly shows the larger abundance of $N_2O$ observed at 31 km (160-200 ppbV) than at 35 km (50-140 ppbV). This indicates the time

needed to transport air masses between the two altitudes, which in turn favours larger $NO_2$ production at 35 km of around 4.5-5.5 ppbV in comparison to 3-4.5 ppbV at 31 km. $NO_2$ produced from the oxidation of $N_2O$ impacts $O_3$. Figure 4b shows the $N_2O$ and $O_3$ correlation. There is a linear relation, as the lifetime of both tracers in this region is greater than their vertical transport timescales (Bönisch et al., 2011). Both panels of Fig. 4 show quite compact correlations between the tracers, which indicate well mixed air masses (Hegglin et al., 2006).

To obtain more detailed information about tracer distributions, in particular on the $NO_2$ impact on the observed negative $O_3$ change, we present $NO_2$-$O_3$ scatter plots at 31.5 km in the tropics in Fig.5 from SCIAMACHY measurements and TOMCAT simulations CNTL, fSG and fDYN. Data points indicate zonal monthly mean values during January 2004-April 2012. Here TOMCAT results are interpolated to the SCIAMACHY vertical grid. Solid lines in each panel specify linear fits to corresponding data points and represent the chemical link between $O_3$ and $NO_2$. All panels of Fig. 5 show the expected negative

correlation of $O_3$ with $NO_2$. The SCIAMACHY $NO_2$-$O_3$ distribution (Fig. 5a) agrees well with TOMCAT CNTL simulation (Fig. 5b), though modelled $NO_2$ and $O_3$ are lower in comparison with measurements.





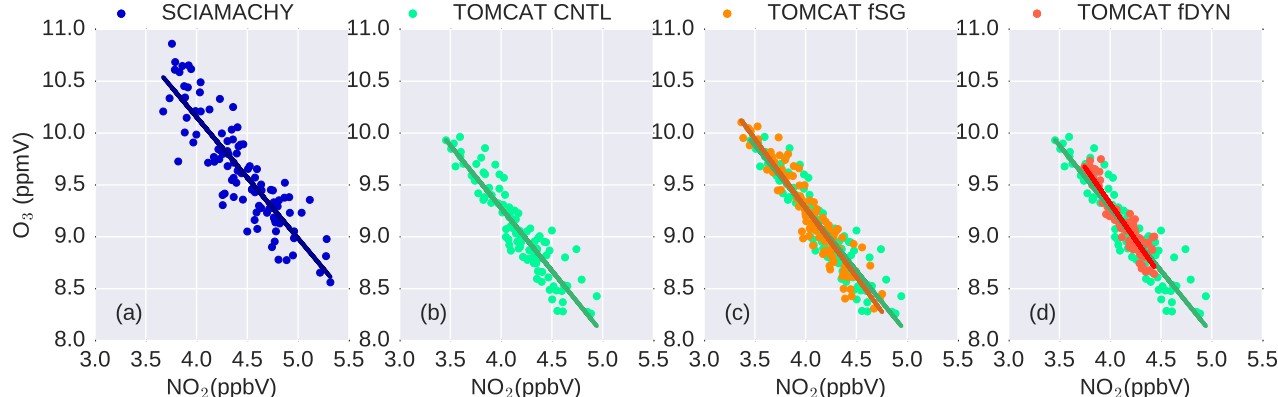

**Figure 5.** $NO_2$-$O_3$ scatter plots in the tropical stratosphere at altitude 31.5 km from (a) SCIAMACHY and TOMCAT simulations (b) CNTL, (c) CNTL and fSG, and (d) CNTL and fDYN. Colour coding denotes data source: SCIAMACHY (dark blue), TOMCAT: CNTL (green), fSG (orange), fDYN (red). Solid lines specify linear fits to the data points.

To further investigate the impact of dynamics on $NO_2$-$O_3$ changes, Fig. 5c shows a combined scatter plot from both simulations CNTL and fSG. In conditions of unchanged tropospheric mixing ratios of source gases (fSG, orange points), the data scatter and slopes do not change significantly in comparison with the control simulation (CNTL, green points). Both simulations are performed with the same dynamical forcing. In contrast, the $NO_2$-$O_3$ scatter from CNTL, and fDYN TOMCAT
simulations (Fig. 5d) differ significantly. In the absence of dynamical changes (red points) the $NO_2$ and $O_3$ scatter do not show such large variability as in run CNTL (green points), which highlights the impact of transport and indicates different tracer distributions with and without dynamical changes. However, the slopes are similar in both simulations, which represents the chemical impact of $NO_x$ changes on $O_3$. Therefore, the $NO_2$-$O_3$ scatter plots from the model calculations confirm the notion that observed $O_3$ changes are linked to $NO_x$ chemistry in the tropical mid-stratosphere. Also, it follows from the different
TOMCAT simulations that these chemical changes on shorter timescales are ultimately driven by dynamical variations.

Recognising the tight relationships within the tropical mid-stratosphere $N_2O$-$NO_x$-$O_3$ chemistry, seen in Figs. 4 and 5, we further calculated correlation coefficients ($R^2$), including the dynamical AoA tracer. Figure 6 shows the correlation heatmap for AoA, $N_2O$, NO, $NO_2$, and $O_3$ for the period January 2004–April 2012 in this region. Repeated information is excluded from the heatmap. The correlations ($R^2$) between the chemical species $N_2O$, NO, $NO_2$, and $O_3$ are very high and in all cases exceed 0.9.
This is consistent with tracer-tracer correlations shown in Figs. 4a,b and 5a-d. The $R^2$ value for $N_2O$-$O_3$ is lower in comparison with that for $NO_2$-$O_3$. This difference is larger when looking at seasonal values (not shown here). Such differences in $R^2$ are explained by the overall regulation of the $O_3$ abundance in the tropical mid-stratosphere. Ozone is mainly destroyed by $NO_x$ in this altitude region and the strong chemical link between $O_3$ and $NO_x$ is confirmed by the high anti-correlation ($R^2$=0.92). A strong anti-correlation is expected between $N_2O$ and $NO_y$ as these are both long-lived tracers in the mid-lower stratosphere



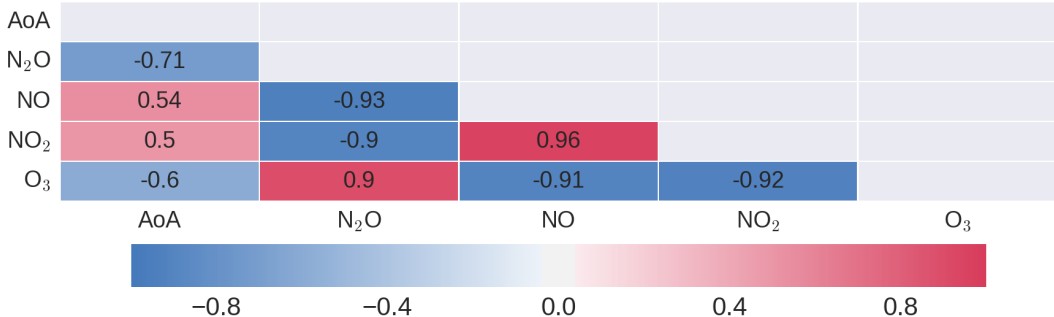

**Figure 6.** Correlation ($R^2$) heatmap for AoA, $N_2O$, NO, $NO_2$, and $O_3$ from TOMCAT CNTL run for the period January 2004–April 2012 in the tropical mid-stratosphere. Repeated information was excluded from the heatmap.

and $N_2O$ is the source of $NO_y$. As the amount of $NO_x$ also scales with the amount of $NO_y$, a fairly strong correlation ($R^2$=0.9) exists between $N_2O$ and $O_3$, even in the mid-stratosphere where the ozone photochemical lifetime becomes short.

The correlation of AoA with all tracers is rather moderate (with absolute values within the range of 0.5-0.71), as transport (or AoA) does not directly control NO, $NO_2$, and $O_3$ in this region. The anti-correlation of $N_2O$ and AoA is also moderate
($R^2$=-0.71), which is an unexpected finding, as the major source of $N_2O$ in the tropical mid-stratosphere is the upwelling from lower altitudes (see Sect. 1).

### 3.3 $N_2O$ - Age of Air relationship

To improve our understanding of the AoA-$N_2O$ relation, Fig. 7 shows profiles of $N_2O$, $N_2O$ loss and $N_2O$ lifetime from the TOMCAT CNTL run, averaged over the period 2004-2012. A significant decrease of $N_2O$ concentrations with altitude is seen
in Fig. 7a, in particular a sharp decrease around 20 km altitude. Figure 7b shows that the largest (∼90%) $N_2O$ loss is caused by photolysis (R7, orange), which starts to become important at around 20 km altitude. About 5% of $N_2O$ reacts with $O(^1D)$ above 26 km, where the concentration of $O(^1D)$ starts slowly increasing due to the reaction (R3a). As the consequence of these $N_2O$ loss reactions, its average lifetime (shown in Fig. 7c), calculated as the ratio of mean $N_2O$ concentration and its total loss is also strongly altitude-dependent. It varies from more than 100 years at 20 km to less than 1 year at 35 km. In particular, in
the altitude range 30-35 km the $N_2O$ lifetime varies by a factor of two (Fig. 7c).

To investigate the link between transport and $N_2O$, we show in Fig. 8 (a) zonally averaged climatological mean AoA (years) and (b) AoA linear changes (years per decade) as a function of latitude in the stratosphere from TOMCAT CNTL simulation during January 2004–April 2012. The dashed rectangle indicates the region of interest in the tropical mid-stratosphere and AoA is shown on native TOMCAT vertical grid. Hatched areas in Fig. 8b show regions where changes are significant at the $2\sigma$
level. In the tropical mid-stratosphere, according to Fig. 8a the average lifetime of air is around 3.5 years, and according Fig. 8b there are no statistically significant changes of AoA. The absence of AoA changes in considered region is on the one hand in agreement with Aschmann et al. (2014), who demonstrated that the deep branch of BDC does not show significant changes.

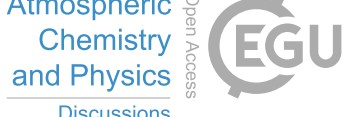

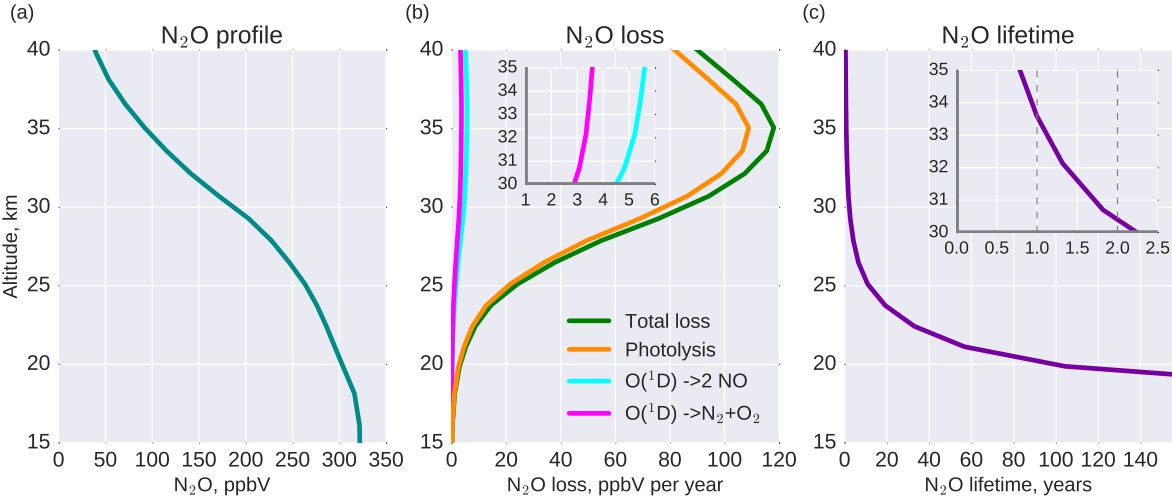

**Figure 7.** Average profiles of (a) $N_2O$ (ppbV), (b) $N_2O$ loss (ppbV per year), and (c) $N_2O$ lifetime (years) from TOMCAT for the period 2004-2012. Colour coding in panel (b) indicates the source of $N_2O$ loss: total loss - green; loss via photolysis (R7) - orange; loss via oxidation with $NO_x$ production (R8a) - turquoise; loss via oxidation without $NO_x$ production (R8b) - magenta.

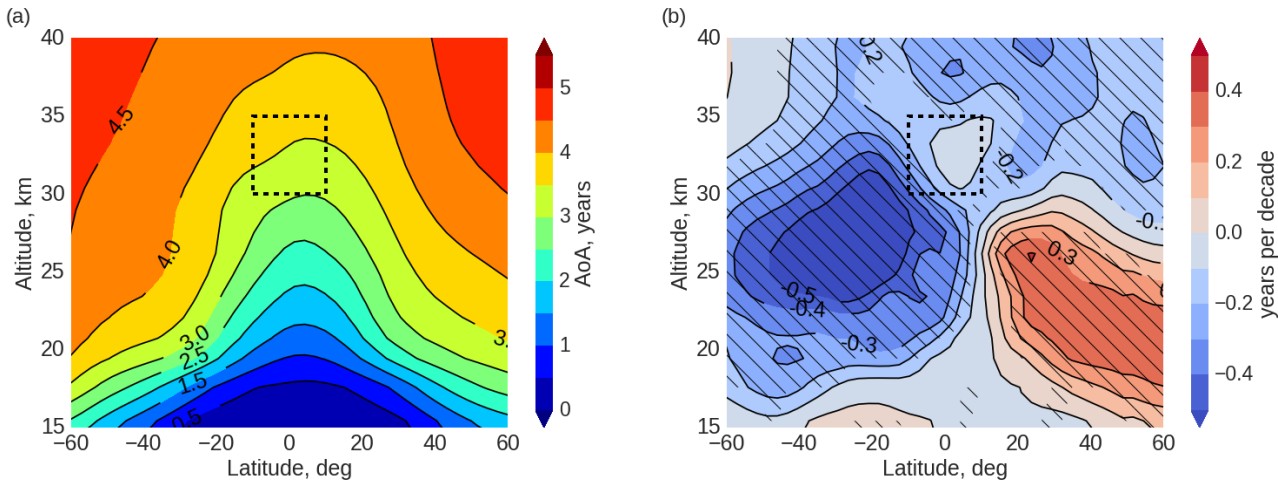

**Figure 8.** Latitude-altitude AoA (a) zonally averaged distribution (years), (b) linear changes (years per decade) from MLR model based on TOMCAT CNTL simulation during January 2004–April 2012: latitude range from $60°S$ to $60°N$, altitude range from 15 to 40 km. The dashed rectangle indicates the region of the tropical mid-stratosphere investigated in this paper. Hatched areas in panel (b) show changes significant at the $2\sigma$ level.

On the other hand, it is apparently inconsistent: 1) with $N_2O$ negative changes identified in the region of interest as shown in Fig. 3g, and 2) with conclusions of Nedoluha et al. (2015b), who suggested a decrease in upwelling speed as a possible reason for the observed $O_3$ decline at 10 hPa (around 30-35 km altitude).



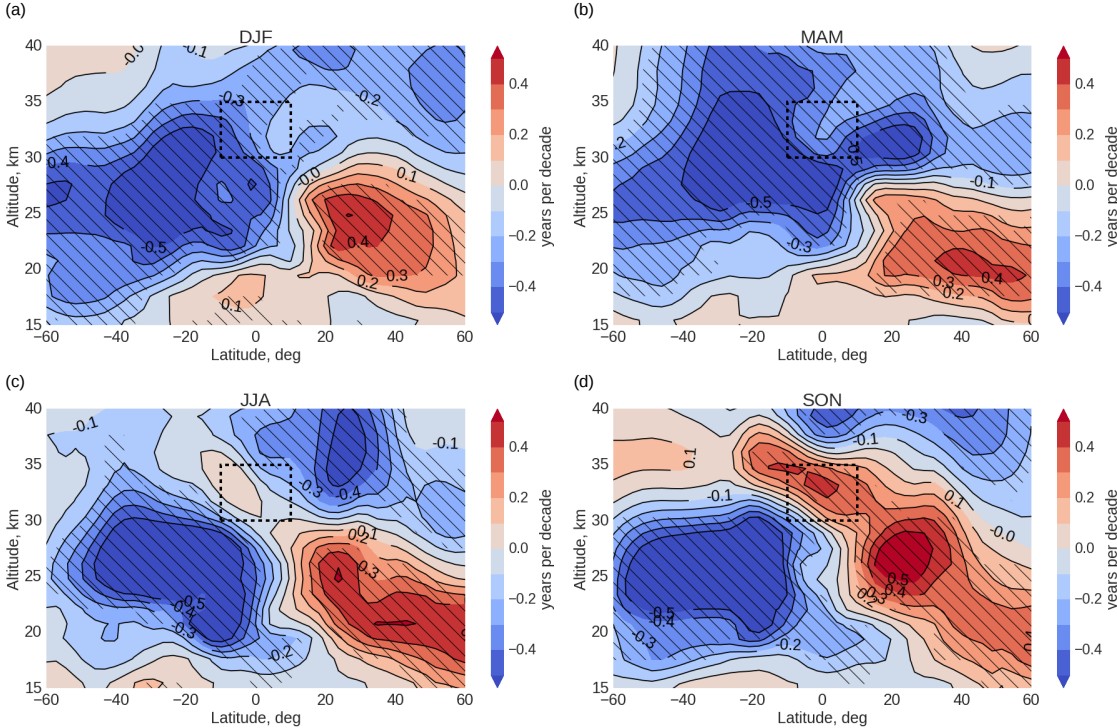

**Figure 9.** Latitude-altitude distribution of AoA changes (years per decade) from MLR model based on the TOMCAT CNTL simulation during January 2004–April 2012 for (a) DJF, (b) MAM, (c) JJA, (d) SON: latitude range from 60°S to 60°N, altitude range from 15 to 40 km. Hatched areas show changes significant at the $2\sigma$ level. The dashed rectangle indicates the region of the tropical mid-stratosphere investigated in this paper.

To further improve our understanding of AoA changes, we show in Fig. 9 seasonal analysis of AoA linear changes (years per decade) from MLR model during January 2004–April 2012 based on TOMCAT CNTL simulation for (a) DJF, (b) March-April-May (MAM), (c) June-July-August (JJA), and (d) September-October-November (SON). Figure 9 shows that in the tropical mid-stratosphere AoA changes vary significantly during seasons: AoA decreases during DJFs and MAMs (Fig. 9a,b) and

5     increases during SONs (Fig. 9d). During JJAs (Fig. 9c) no statistically significant changes of AoA in tropical mid-stratosphere were identified. Observed seasonality in AoA changes in the tropical mid-stratosphere leads to insignificant changes when averaged over the entire year (seen in Fig. 8b). Another interesting pattern shown in Fig. 8b is the clear asymmetry between the hemispheres, with negative AoA changes in the southern and positive AoA changes in the northern hemispheres. This asymmetry is consistent with the results presented in Sect. 3.1 for $N_2O$ changes as the long-lived tracer (Fig. 3g,h) and in

10     agreement with Mahieu et al. (2014) and Haenel et al. (2015). The hemispheric asymmetry, however, remains unchanged within all seasons (Fig. 9a-c).



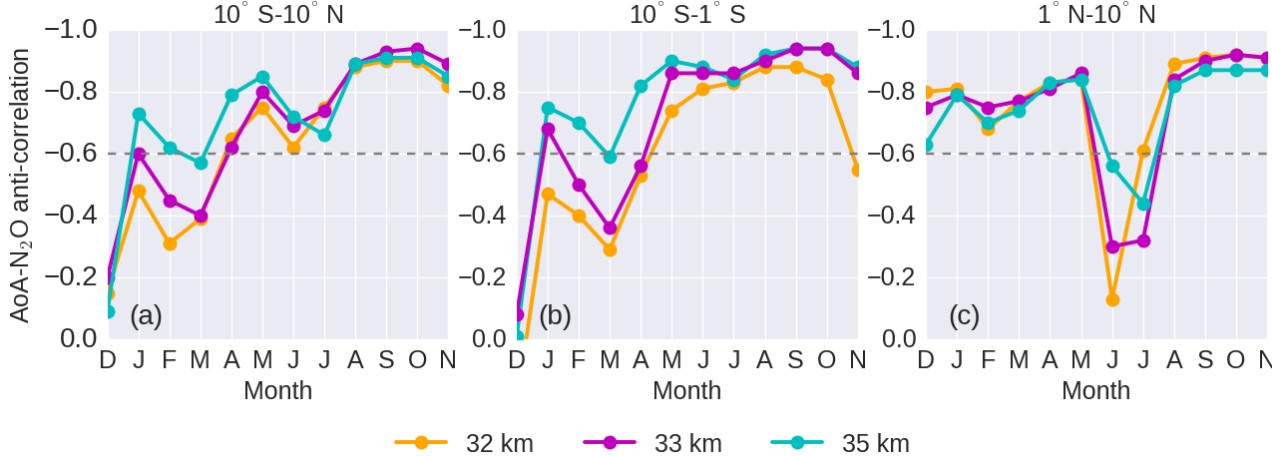

**Figure 10.** $N_2O$-AoA anti-correlation ($R^2$) as a function of month averaged for the period January 2004–April 2012 for (a) $10°S$-$10°N$, (b) $10°S$-$1°S$, and (c) $1°N$-$10°N$. Colour coding indicates altitude: 32 km (orange), 33 km (magenta), and 35 km (cyan). The dashed horizontal lines indicate the lower edge of moderate correlation, which was selected to be $R^2$=0.6.

From the above, the major resulting questions is: how are $N_2O$ and transport (via AoA) connected? To answer this, we analyse $N_2O$-AoA correlation coefficients as a function of month (Fig. 10) at altitudes 32, 33, and 35 km. To overcome any hemispheric dependencies, we split the tropics into southern ($10°$-$1°S$) and northern ($1°$-$10°N$) regions. Correlation coefficients were calculated on the native TOMCAT grid. Horizontal dashed lines indicate the lower edge of moderate correlation, which

is represented by the absolute value of $R^2$=0.6.

Figure 10a shows that in the tropical region ($10°S$-$10°N$) the AoA-$N_2O$ anti-correlation is very low during December-March. During the other months of the year, it is moderate and reaches maximum values (around 0.9) during late NH summer (August) and autumn (September, October) months. Very similar seasonal behaviour is also observed in the tropical region of the southern hemisphere (Fig. 10b) with the minimum correlation occuring during December-March (southern hemisphere

summer) and maximum during May-October (southern hemisphere winter). In contrast, in the tropical region of the northern hemisphere (Fig. 10c) a significant decrease in AoA-$N_2O$ anti-correlation is observed during summer months (June-July). Similar seasonal variations of $N_2O$-AoA anti-correlation are observed in narrower latitude bands ($4°S$-$4°N$) which are shown in the Supplements Fig. S3. The common characteristic of seasonal changes of AoA-$N_2O$ is that a significant decrease of the anti-correlation is observed during local summer in each hemisphere. This is the period when the strength of the BDC is the

lowest (Kodama et al., 2007, and references therein) and no significant changes in AoA are observed. The overall correlation for inner tropics from $10°S$ to $10°N$, as shown in Fig. 10a, combines the behaviour of both hemispheres.

With knowledge of the existence of strong seasonal dependencies in AoA variability, and therefore in $N_2O$, we have analysed the AoA-$N_2O$ relation as a function of month, averaged for the period January 2004-April 2012. Figure 11 shows $N_2O$ mixing ratio and AoA averaged over January 2004-April 2012 as a function of month and altitude. The matching of the colour and




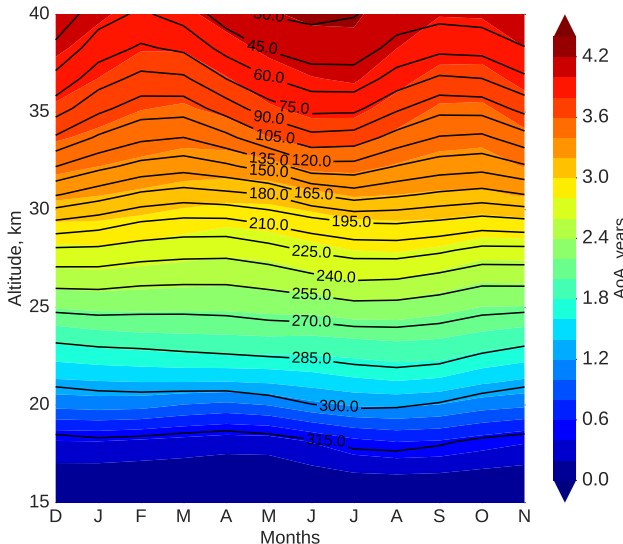

**Figure 11.** Annual cycle of monthly mean tropical $N_2O$ (ppbV, contours, 15 ppbV interval) and AoA (years, colours, 0.2 yr interval) as a function of altitude from TOMCAT run CNTL, averaged over the period January 2004–April 2012.

contour isolines is pronounced, confirming a direct link between $N_2O$ and AoA. Furthermore, as $N_2O$ is transported from the troposphere, its concentrations decrease with altitude (see also Sect. 1). In the lower stratosphere (15-20 km) the seasonal variations of $N_2O$ and AoA exist, but are not as pronounced as in the mid-stratosphere (30-35 km). Moreover, the seasonal variations in $N_2O$ are larger than the seasonal variations in AoA, so the correlation breaks down (as seen from Fig. 8). There are two distinct seasonal features seen in the $N_2O$-AoA distribution in the mid-stratosphere, which increase at a given altitude: during January-March and September-November. During these periods AoA becomes lower in comparison with the rest of the year (indicating younger air) and therefore more $N_2O$ is transported to these altitudes.

### 3.4 Observed changes in the tropical mid-stratosphere

Figures 10 and 11 show the seasonal variations of AoA and $N_2O$ in the tropical mid-stratosphere. To further investigate the possible chemical impact on other species, we analysed linear changes of AoA, $N_2O$, $NO_2$, and $O_3$ from TOMCAT run CNTL and SCIAMACHY measurements for each calendar month (see Supplements Fig. S4-S7). TOMCAT run CNTL in general shows lower $O_3$ and $NO_2$ concentrations compared to SCIAMACHY measurements. The underestimation of modelled $NO_2$ and $O_3$ is also evident when comparing Figures 5a and 5b, but the slope of the modelled anti-correlation regression line agrees very well with that of the SCIAMACHY observations. The reason for these biases between model and measurements is not clear. However, if the model $NO_2$ increases then $O_3$ will decrease even further (Fig. 5). Therefore, it is unlikely that transport errors are the cause. Either the model underestimates the production of $O_3$ from $O_2$ photolysis in this region, or there are



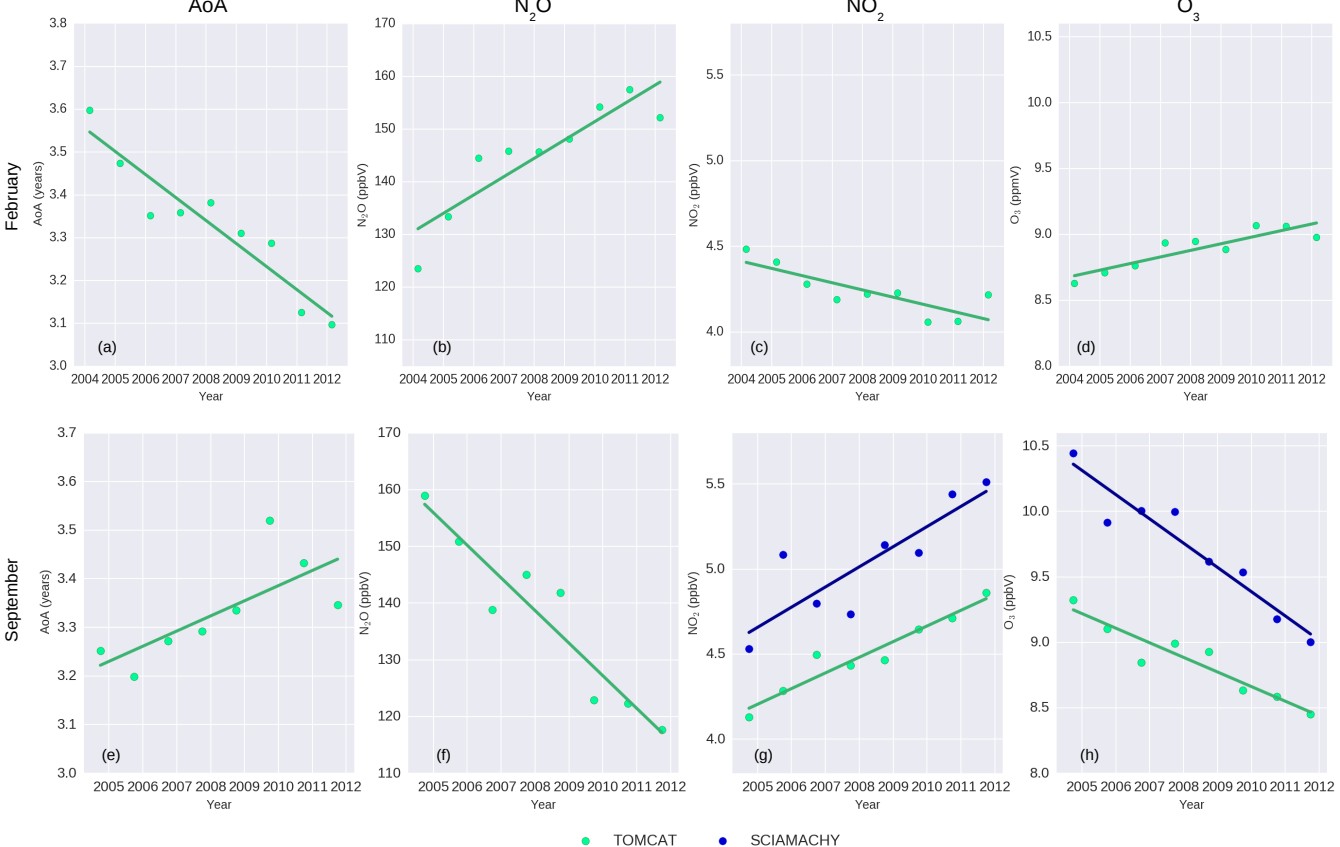

**Figure 12.** Linear changes of AoA, $N_2O$, $NO_2$, and $O_3$ minus QBO effect averaged over (a-d) Februaries 2004-2012 and (e-h) Septembers 2004-2011 in the tropical stratosphere between 30-35 km altitude. Colour coding indicates the data source: TOMCAT CNTL simulation (green), and SCIAMACHY measurements (dark blue). There are no significant changes in SCIAMACHY measurements taken in February (see Supplements Fig. S4), therefore they are excluded from the figure.

uncertainties in the model $NO_x$ chemistry which means the impact of $NO_2$ on $O_3$ is less than modelled. The latter uncertainties would then be associated with the reactions $O + NO_2$, $NO + O_3$, or $NO_2$ photolysis.

The upper panels of Fig. 12 show statistically significant linear changes of February AoA, $N_2O$, $NO_2$, and $O_3$ from TOM-CAT CNTL run. The Februray decrease of AoA led to less intense $O_3$ destruction. Similar results are observed for January
5 (Supplements Fig. S4). In particular, the faster upwelling as indicated by the decrease in AoA (Fig. 12a) results in more intense $N_2O$ transport and smaller photolytic destruction, therefore $N_2O$ increases with time (Fig. 12b) while $NO_2$ decreases (Fig. 12c) due to shorter residence time of $N_2O$, i.e. there is less time to produce $NO_x$ species via the $O(^1D)$ reaction (R8a). Finally, a slight increase of $O_3$ is observed in the tropical mid-stratosphere (Fig. 12d). SCIAMACHY measurements do not show any




significant changes of $NO_2$ and $O_3$ during Januaries and Februaries (see Supplements Fig. S4). Therefore, we excluded these measurements from panels c and d in Fig. 12. We found that SCIAMACHY linear changes showed larger errors in comparison with the TOMCAT model. Our analysis showed that TOMCAT changes would also become insignificant if TOMCAT had the same errors as SCIAMACHY measurements. The larger errors of SCIAMACHY changes can be explained by the short

analysis period and limitations of satellite measurements in tropics.

The lower panels of Fig. 12 present linear changes of September AoA, $N_2O$, $NO_2$, and $O_3$, where all changes are opposite to the winter months shown in the upper panels. The variations for October are similar to September (see Supplements Fig. S7). Positive AoA changes (Fig. 12e) indicate a significant transport slow-down or additional air mixing. As a result, there is less $N_2O$ transported from the troposphere (Fig. 12f) and more $N_2O$ is photolytically destroyed. The residence time in the tropical

mid-stratosphere gets longer, producing more $NO_2$ (Fig. 12g) which destroy $O_3$ more effectively (Fig. 12h). SCIAMACHY $NO_2$ and $O_3$ changes agree well with modelled data in September and October (panels g, h in Fig. 12 and Supplements Fig. S7).

Thus, negative AoA changes during boreal winter months (January and February) and positive AoA changes during boreal autumn months (September and October) cancel out and do not show any significant changes when averaged over the entire

year (Fig. 8). However, the chemical responses of $N_2O$, $NO_2$ and as consequence $O_3$ do not cancel out in a yearly average. This effect occurs as a result of a non-linear relation between AoA and $N_2O$ and can be explained as follows. In the absence of the photolytic loss, the increase/decrease of the stratospheric $N_2O$ is controlled only by the increase/decrease of the upwelling speed. In turn, the photolytic loss of $N_2O$ is determined by its residence time in the stratosphere, with shorter residence time (i.e. higher upwelling speed) resulting in higher $N_2O$ amounts and vice versa. As the overall amount of $N_2O$ is controlled by

both transport and photolysis, its changes do not cancel in the yearly average as opposed to AoA, which almost solely depends on the speed of BDC (i.e. tropical upwelling). Because of a strong chemical relation between $N_2O$ and both $NO_2$ and $O_3$, their seasonal behaviour is determined by that of $N_2O$.

## 4   Conclusions

We have analysed $O_3$ changes in the tropical mid-stratosphere during January 2004-April 2012 as observed by SCIAMACHY

and simulated by the TOMCAT CTM. We find that the model, forced by ECMWF reanalyses, captures well the observed linear $O_3$ changes within the analysed period. Using a set of TOMCAT simulation with different dynamical and chemical forcings we showed that the decline in $O_3$ is ultimately dynamically controlled and occurs due to increases of $NO_2$, which then chemically removes $O_3$. The $NO_2$ increases are due to a longer residence time of its main source $N_2O$, which is long-lived so changes in its abundance indicate variations in the tropical upwelling. These results are in agreement with finding of Nedoluha et al.

(2015b). To further investigate whether there was a decrease of tropical upwelling we analysed the AoA from the TOMCAT model. However, the AoA simulations did not show any significant annual mean changes in the tropical mid-stratosphere, in apparent contradiction with conclusions of Nedoluha et al. (2015b).



With the knowledge of dynamically driven $N_2O$-$NO_2$-$O_3$ changes but no significant changes of mean AoA, we performed a detailed analysis of linear changes for each month separately within the period January 2004-April 2012. We find that during boreal autumn months, i.e. September and October in the north, there is a significant transport slow-down or additional air mixing which corresponds to positive changes of AoA. These positive changes cause longer residence time of $N_2O$, leading to increased $NO_x$ production and stronger $O_3$ loss. SCIAMACHY and TOMCAT $O_3$ and $NO_2$ changes are consistent in that regard. In contrast, we find that during boreal winter months, i.e. January and February, the AoA simulations show a transport speed-up. This decreases the residence time of $N_2O$, so less $NO_x$ is produced and consequently less $O_3$ is destroyed. While the TOMCAT model shows significant $NO_x$ decrease and $O_3$ increase, the SCIAMACHY changes are not significant during these winter months. This is associated with larger errors of the linear regression in the satellite data.

Starting from the seasonal variation of AoA changes and its impact on annual mean trends in the tropical mid-stratosphere as presented in this paper, some questions still remain and should be the subject of further studies. Is the shift of subtropical transport barriers, suggested by Eckert et al. (2014) and Stiller et al. (2017) linked to the seasonal AoA changes observed here? Or, is this a result of the different behaviour of the shallow and deep branches of the BDC, i.e. hiatus in the acceleration of the shallow branch, strengthening of the transition branch and no significant changes in the deep branch (Aschmann et al., 2014)? The cooling of Eastern Pacific (Meehl et al., 2011) could also affect $O_3$ changes via upwelling, although our analysis of sea surface temperatures (not shown here) in Eastern Pacific did not show significant monthly variations. Another plausible explanation of changes in the transport regime could be from contribution of the planetary wave forcing (Chen and Sun, 2011), as we find that a significant decrease in the $N_2O$-AoA correlations occurs during local summers, when the wave activity and therefore the strength of the upwelling is the lowest. In particular, the impact of variations in the wave activity on seasonal build-up of $O_3$ was also described by Shepherd (2007). However, all of these possible explanations require additional investigation to decide which processes dominate.

Overall, the non-linear relation of AoA and $N_2O$ and their month-to-month changes presented in this paper explain well the observed $O_3$ decline in the tropical mid-stratosphere. With the application of a detailed CTM we are able not only to confirm the $O_3$ decline, but also describe chemical impacts and define the role of dynamics on the observed changes. Having identified in this study the impact of a seasonal dependency of the upwelling speed on the tropical mid-stratospheric $O_3$, a better understanding of the possible drivers of this behaviour is now required. However, the CTM with its specified meteorology cannot be used to determine the main drivers of the dynamical changes. Consequently, the application of interactive dynamical models is needed. The interpretation of the observed changes will give us an understanding whether $O_3$ decline in the tropical mid-stratosphere is a part of natural variability, human impact, or a complex interaction of both factors.

*Data availability.*

SCIAMACHY $O_3$ and $NO_2$ data are available after registration at http://www.iup.uni-bremen.de/scia-arc/. Results of TOM-CAT simulations are available upon request from the authors. QBO equatorial winds at 10 and 30 hPa were taken from http://www.geo.fu-berlin.de/en/met/ag/strat/produkte/qbo/index.html. The anomalies of the Nino 3.4 index were downloaded

©c Author(s) 2018. CC BY 4.0 License.





from http://www.cpc.ncep.noaa.gov/data/indices/. Data of Mg II index from GOME, SCIAMACHY, and GOME-2 were taken from http://www.iup.uni-bremen.de/UVSAT/Datasets/mgii.

*Code and data availability.*

*Competing interests.*

5    The authors declare that they have no conflict of interest.

*Disclaimer.*

*Acknowledgements.* This research has been partly funded by the University and State of Bremen, by the Postgraduate International Programme in Physics and Electrical Engineering (PIP) of the University of Bremen, by the project SHARP-II-OCF, and by a BremenIDEA out Scholarship, promoted by the German Academic Exchange Service (DAAD) and funded by the Federal Ministry of Education and Research
10   (BMBF). The model simulations were performed on the UK national Archer and Leeds ARC HPC facilities. The authors thank A. Maycock and A. Chrysanthou for comments on early stages of this research. The data presented were partially obtained using the German High Performance Computer Center North (HLRN) service, whose support is gratefully acknowledged.





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
