# Peer review of "Dynamically controlled ozone decline in the tropical mid-stratosphere observed by SCIAMACHY"

_Atmospheric Chemistry and Physics, 2018_

## Referee Comment (RC1) · Anonymous Referee #1 · 17 Sep 2018

This manuscript addresses the trend in N2O, and the resulting trend in O3, which has been observed in the tropical mid-stratosphere (30-35km) on decadal scales by several instruments. The overall SCIAMACHY measurements included here also show this trend in O3, and the trend in NO2 which one expects from the dynamical changes which drive the N2O trend.

The significant contribution that this manuscript makes, is to show that, according to TOMCAT simulations, there is (from 2004-2012) an increase in Age-of-Air (AoA) in the tropical mid-stratosphere (30-35km) during some seasons, and a decrease in AoA during others. This result seems plausible, and offers the interesting possibility of

changing N2O (and hence O3) in this region, while perhaps not changing AoA as much as might otherwise be expected.

While this is quite interesting, the authors have somewhat oversold the conclusion. They can conclude from their model that there is "no statistically significant trend in AoA", but they cannot say that there is "no change in AoA" (in fact, there is a small overall increase in AoA in their model results). While I have no reason to doubt the model results, their explanations for why the seasonal variation in AoA causes N2O and AoA change differently do not provide any useful insight. It is, of course, highly desirable to have a better understanding of the N2O and AoA relationship, but unless the explanations can be greatly improved I would recommend dropping these from the manuscript.

I also have some serious concerns with the presentation of the SCIAMACHY measurements in the manuscript. The authors need to make very clear to the reader that, contrary to the model, they have not found any SCIAMACHY data which shows statistically significant increase in SCIAMACHY O3, or a decrease in NO2, during any particular month or season. It is certainly not appropriate that the measurements during the months when the model says that an increase in O3 or a decrease in NO2 should occur, and which shows no significant measurement trend, are relegated to the supplement, while at the same time the data during months when the opposite trends occur and the model and measurement trends agree (at least in sign and significance) are shown alongside the model in the main text.

More detailed comments (some of which repeat points from above):

Page 6 line 19 – "Global coverage of SCIAMACHY limb measurements was obtained within 6 days at the equator and less elsewhere." It's not clear to me what this means. Perhaps the authors are requiring some maximum distance between measurements. Unless the authors wish to provide a clear definition I would recommend dropping this sentence.

Page 6 line 24 – "the errors of single measurements are mostly normally distributed and no additional issues with outliers have been reported." I think this means that there was no need to remove outliers, but if this is the case please say this more clearly. If this is not the case then please rewrite the sentence to better explain what is meant.

Page 6 line 25 – "Consequently, we assumed that the random errors of zonal monthly means could be neglected." Without knowing at this point how you are using the data it is hard to know whether this is reasonable or not. I would drop this sentence from here and perhaps make the point.

Page 7 – "In the latitudes between 50-60N and within altitude range 15-26 km we applied cumulative eddy heat flux instead of harmonic fit terms. We used ERA-Interim eddy heat flux at 50 hPa integrated from 45N to 75N with the time lag of 2 months." I am not acquainted with this method. Do other groups do this? Is there a reference? If not, please give some explanation/justification.

Page 13 line 21 –"The absence of AOA changes in considered region . . ." This is a fundamental conclusion of the paper, but it represents an unjustified conclusion from the statistics. One cannot conclude from the absence of statistical significance that "there is no change in AOA". One can only conclude that "there is no statistically significant trend".

Figure 9 is particularly interesting.

Figure 12 – "There are no significant changes in SCIAMACHY measurements taken in February (see Supplements Fig. S4), therefore they are excluded from the figure." One can't simply include the SCIAMACHY measurements for a particular month per year when they fit the model, and then ignore them when they don't. The SCIAMACHY NO2 results as shown in the supplement are almost significant at the 2-sigma level (they are certainly significant at 1-sigma) and are in the opposite direction of what the model shows. The easiest solution would be for the authors to conclude that the SCIAMACHY measurements, when plotted as one month per year, simply aren't up to this, and

therefore need to be dropped from this figure entirely. The SCIAMACHY results as shown in Figure 2 and 5 certainly do demonstrate the value of this measurements when they are not subsampled as in Figure 12.

Page 19 line 13- This paragraph purports to explain the absence of change in AoA. While it is certainly possible that one could have a change in N2O and not a change in AoA, this point has not been proven. At the same time, the explanation seems to be simply a complicated statement of the fact that changes in N2O are governed by changes in upwelling speed, which obviously couple to AoA. Unless the authors can offer some additional insight here I would recommend dropping this paragraph.

Supplement – The notation of SCIAMACHY, TOMCAT, and Insignificant is confusing, since the gray Insignificant lines can be either of the former two. The current notation obscures the important fact that the subdivided SCIAMACHY measurements never show a significant trend in the opposite direction to the overall trend in N2O and O3. I recommend using just green, blue, and, if the authors think it is helpful, a dotted version of these colored lines for an insignificant trend.

---

## Referee Comment (RC2) · Anonymous Referee #2 · 28 Sep 2018

The authors aimed to understand the negative ozone change seen in the middle tropical stratosphere, and in doing so made the link that increases in NO2 as a result of dynamical changes were causing the loss of ozone in the region of focus around 30-35 km. However, they were not able to link this to a statistically significant change in the age of air, which is also an interesting result. Nevertheless, the importance of understanding multiple chemical and dynamical drivers in the stratosphere is highlighted and the authors present interesting results and raise questions worth investigating further.

However, my concern is that some of the points made, and hypotheses, are not well supported by what is presented, or the authors are not explicit and careful with how

they present results (e.g. correlation coefficient, below). I think this work is useful, and should be published, but changes are needed to make it explicitly clear what (i) can definitely be said from the observations, model and comparisons, (ii) what are the hypotheses the authors are putting forward, and (iii) what are the clear open questions that need to be addressed in future.

Comments: 1. I am in agreement with the other referee that the non-significant, even opposite signal (and sign of trend) in February, though non-significant (in the supplement), is not addressed head on. Data is often messy and difficult to deal with especially when comparing with a model, and should be presented front and centre even if there is a contradiction or lack of evidence to contend with. This actually requires a deeper discussion, because if the model disagrees with the data in the sign of the trend (and it appears consistent between NO2 and O3 in February in the supplementary materials despite the non-significance) then that raises questions that need to be highlighted (for example, is it a model or an observational problem?). I won't labour on this point further, or repeat points raised by the other referee, as the other referee has spent quite some time on points related to this.

2. Page 4, L20-23: is this relationship specifically in the 30-35 km tropical region of the study (see comment 2 below).

3. Page 4, L26: actually I would argue that the decrease Kyrola et al., 2013 found was up to 6-8% at its core (Fig. 16), which is more in line with that quoted for Gebhardt et al 2014. However, the core of the negative region in Gebhardt et al., 2014 is upward of -18% (Fig. 8). Could the authors be clear in what they mean here since I believe the -10% refers to the 20S-20N (Fig 7) profile; since the authors focus in on +/-10 deg. latitude region, the higher value seems more appropriate but then the estimate in this manuscript is almost 2x smaller. I assume, though perhaps the authors should check, this difference is due to a different time period and set of regressors used? At the very least please be explicit about what region the numbers represent and are comparable to the region focused on in this manuscript.

4.  Page 8, L4-9: I am not sure I agree that it is consistent to ignore the monthly autocorrelation when using all months. It seems to me consistent not to use it for single-month (i.e. Jan only, etc) estimates (since there should be no autocorrelation between the months 12 months apart) and to indeed consider autocorrelation for the full time series since that is typically the case if they are next to each other in a continuous timeseries. This is only reasonable if you can state explicitly that there is no change in the significance - does considering it have an effect on your conclusions?

5.  Page 9, L14: the inference the authors make from Fig.3 is that chemistry has little impact on the 30-35 km tropical region; for O3 and N2O I think this is reasonable. But for 3d-f it seems that in the box, NO2 is roughly split 70/30 or maybe 50/50 in the peak positive change. So it isn't clear to me if this statement is fully backed up by the plot (or perhaps its a non-linear interaction?). Please could the authors comment on this, perhaps with values.

6.  While Fig 4a. shows a combined non-linear shape, it appears that the anti-correlation (linear slope for each level) reduces with higher altitude, being almost flat at 35 km (green). Why does this happen? Does this indicate that the mechanism proposed is no longer operating as efficiently in the upper part of the box?

7.  Fig 6, 10, and all discussion related to the $R^2$ statistic: this is very confusing and needs to be stated explicitly and correctly. $R^2$ is formally the "coefficient of determination", which can be the square of, but not same as the "correlation coefficient". Further $R^2$ can only range from 0 to 1, while the correlation coefficient can range from -1 to +1. Please check all instances of this and be correct in its usage; in many places this is confusing and leads the reader to have to try and work out what the authors mean.

8.  Page 16, L5: 0.6 is an arbitrary threshold; please state this explicitly.

9.  Fig. 11: is this also integrated over 10S-10N?

10.  Page 19, L4-5: Is this a hypothesis or a demonstrable fact? I do not understand

why it is a limitation of the measurements, given the description earlier of the limb observations being well-distributed in the tropics and the period being considered is the same for the model data. If the effect is demonstrable, then this would provide good evidence the model is correct and why we don't have to worry about the insignificance and/or inverse correlations. If it is a hypothesis, please state explicitly this is the case.

11. Page 19, L16-22. I'm afraid I found this explanation difficult to follow. Please rewrite to be clearer. Is the summary that the N2O "changes do not cancel in the yearly average" because photolysis has an affect that AoA is not impacted by?

––––––––––––––––––––––––––––

---

## Short Comment (SC1) · 12 Oct 2018

I do not have much to add to the reviewers comments, except the following:

Their discussion on lines 7-11 on page 5 reads as if they are contradicting themselves. Thus line 7 says "decrease in N2O" while lines 8-10 discuss an increase in upwelling leading to "lower N2O oxidation" which necessarily would produce an increase in N2O. It is true that the specific model perturbation we introduced (Nedoluha et al., 2015b) had an increase in upwelling; however, the model-to-model comparison we made we was to show that upwelling strength varies directly as N2O and and inversely as NOy. And the objective was to explain the lower ozone, which would result from weaker

upwelling. I would therefore like to suggest a wording change to be clearer:

Using a 2D chemical-dynamical model, they showed that changes to the tropical upwelling could lead to changes in the N2O oxidation via (R8a) and thus affect the NOy production. Based on this, Nedoluha et al. (2015b) concluded that weaker tropical upwelling could therefore explain the decrease of O3 in the tropical mid-stratosphere.

David Siskind

---

## Author Comment (AC1) · 15 Nov 2018

The authors thank the Referee for his/her thorough reviewing of the manuscript. We address the Referee's criticisms and highlight our proposed improvements below one-by-one in blue. We use the following notation: **P1 L10** means Page 1, Line 10.

This manuscript addresses the trend in N2O, and the resulting trend in O3, which has been observed in the tropical mid-stratosphere (30-35 km) on decadal scales by several instruments. The overall SCIAMACHY measurements included here also show this trend in O3, and the trend in NO2 which one expects from the dynamical changes which drive the N2O trend.

The significant contribution that this manuscript makes, is to show that, according to TOMCAT simulations, there is (from 2004-2012) an increase in Age-of-Air (AoA) in the tropical mid-stratosphere (30-35 km) during some seasons, and a decrease in AoA during others. This result seems plausible, and offers the interesting possibility of changing N2O (and hence O3) in this region, while perhaps not changing AoA as much as might otherwise be expected.

While this is quite interesting, the authors have somewhat oversold the conclusion. They can conclude from their model that there is "no statistically significant trend in AoA", but they cannot say that there is "no change in AoA" (in fact, there is a small overall increase in AoA in their model results).

The Referee is correct in the assertion that the absence of significance in annual mean AoA change does not mean the absence of changes in annual mean $O_3$. However, we discovered that seasonal changes in AoA are significant and they lead to the non-linearity of the physical-chemical mechanisms controlling the $O_3$ amount and distribution.
To address this issue we improved the following formulations in the revised manuscript:
- **P8 L7** we replaced 'The significance of observed changes...' with 'The statistical significance of observed changes...'
- **P14 L1** we replaced 'The absence of AoA changes…' with 'The absence of statistically significant AoA changes..'.

However, we disagree with the Referee that our model results show a small overall increase in AoA. In the area of tropical mid-stratosphere, defined in the manuscript on **P3 L5-6** (10°S-10°N, 30-35 km altitude) the negative AoA changes are statistically insignificant (see Fig. 8b). The small region at ~30-32 km altitude and ~10°S in this box exhibit statistically significant negative changes.

While I have no reason to doubt the model results, their explanations for why the seasonal variation in AoA causes N2O and AoA change differently do not provide any useful insight. It is, of course, highly desirable to have a better understanding of the N2O and AoA relationship, but unless the explanations can be greatly improved I would recommend dropping these from the manuscript.

To address the issue we have rewritten the explanation of the $N_2O$-AoA non-linearity. Please, see below our improvements in the 'More detailed comments' and/or **P19 L19-33.**

I also have some serious concerns with the presentation of the SCIAMACHY measurements in the manuscript. The authors need to make very clear to the reader that, c**ontrary to the model, they have not found any SCIAMACHY data which shows statistically significant increase in SCIAMACHY O3, or a decrease in NO2, during any particular month or season**. It is certainly not appropriate that the measurements during the months when the model says that an increase in O3 or a decrease in NO2 should occur, and which shows no significant measurement trend, are relegated to the supplement, while at the same time the data during months when the opposite trends occur and the model and measurement trends agree (at least in sign and significance) are shown alongside the model in the main text.

The Referee's criticism implies that we have inadequately explained the mechanism which we think explains the behaviour. To address the issues, we have improved the presentation of SCIAMACHY measurements in the manuscript, Specifically, we added SCIAMACHY $NO_2$ and $O_3$ data, which showed insignificant gradients/changes, to Fig. 12c,d and we depicted statistically significant (2-sigma) changes as solid lines, and otherwise as dashed lines.

We also rewrote the explanations related to Fig. 12:

- We mention that SCIAMACHY measurements do not yield statistically significant gradients for the time series of Januaries and Februaries in **P19 L2-3**: 'SCIAMACHY measurements show statistically insignificant changes of $NO_2$ and $O_3$ during Januaries and Februaries (Fig. 12c,d, Supplements Fig. S4)'.
- We also added that contrary to model simulations, SCIAMACHY measurements do not show a $NO_2$ decrease and an $O_3$ increase when analysing changes for any particular calendar month (**P19 L3-5)**: "Contrary to the TOMCAT simulations, SCIAMACHY measurements do not show a statistically significant $NO_2$ decrease and $O_3$ increase when analysing changes for any particular calendar month'.
- We also discuss possible reasons for the model-measurements differences (Fig. 12c,d) on **P19 L5-11**: "From September to February, the gradient of $O_3$ time series increases, becoming more positive for both SCIAMACHY and TOMCAT data, resulting for February in small, statistically insignificant negative gradients for SCIAMACHY observations and small but statistically significant positive gradients for TOMCAT. Similarly for $NO_2$ mixing ratios, from September to February the gradients decrease i.e. they become more positive for both, SCIAMACHY and TOMCAT results. The SCIAMACHY data show larger errors on gradients of the time series for individual months, than those of the TOMCAT model. This results from the stronger oscillating structure in the SCIAMACHY time series. The reasons for the observed oscillations and their strength are not yet unambiguously identified and are under investigation."

More detailed comments (some of which repeat points from above):

Page 6 line 19 – "Global coverage of SCIAMACHY limb measurements was obtained within 6 days at the equator and less elsewhere." It's not clear to me what this means. Perhaps the authors are requiring some maximum distance between measurements. Unless the authors wish to provide a clear definition I would recommend dropping this sentence.
We simplified the sentence on **P6 L19-20** in the revised manuscript as follows: 'For the SCIAMACHY limb measurements, the global coverage was obtained within 6 days.'

Page 6 line 24 – "the errors of single measurements are mostly normally distributed and no additional issues with outliers have been reported." I think this means that there was no need to remove outliers, but if this is the case please say this more clearly. If this is not the case then please rewrite the sentence to better explain what is meant.
We have reworked the sentence and added the reference to Gebhardt et al. (2014) on **P6 L24-26** as follows: 'We calculate zonal monthly mean $O_3$ and $NO_2$ values as arithmetic means as according to Gebhardt et al. (2014) 'the errors of single measurements are expected to be normally distributed and no issue with outliers is known''.

Page 6 line 25 – "Consequently, we assumed that the random errors of zonal monthly means could be neglected." Without knowing at this point how you are using the data it is hard to know whether this is reasonable or not. I would drop this sentence from here and perhaps make the point.
We have withdrawn the sentence. Thank you.

Page 7 – "In the latitudes between 50-60N and within altitude range 15-26 km we applied cumulative eddy heat flux instead of harmonic fit terms. We used ERA-Interim eddy heat flux at 50 hPa integrated from 45N to 75N with the time lag of 2 months." I am not acquainted with this method. Do other groups do this? Is there a reference? If not, please give some explanation/justification.
This method was previously applied by Gebhardt et al. (2014). We improved the sentence by adding the reference to the method on **P7 L25-29** as follows: 'At latitudes between 50-60° N and in the altitude range 15-26 km the cumulative eddy heat flux replaced the harmonic fit terms, similar to Gebhardt et al. (2014). The eddy heat flux was used as a proxy for the transport of stratospheric species due to variation in planetary wave forcing (Dhomse et al.,2006; Weber et al., 2011). Here, we used ERA-Interim eddy heat flux at 50 hPa integrated from 45° N to 75° N with a time lag of 2 months'.

Page 13 line 21 –"The absence of AOA changes in the considered region . . ." This is a fundamental conclusion of the paper, but it represents an unjustified conclusion from the statistics. One cannot conclude from the absence of statistical significance that "there is no change in AOA". One can only conclude that "there is no statistically significant trend".
We agree with the Referee and we reworked the sentences on **P14 L1-2** as follows: '...and according to Fig. 8b there are no statistically significant changes in AoA in the same region. The absence of statistically significant AoA changes here is on the one hand in agreement with … '

Figure 9 is particularly interesting.

Thank you.

Figure 12 – "There are no significant changes in SCIAMACHY measurements taken in February (see Supplements Fig. S4), therefore they are excluded from the figure." One can't simply include the SCIAMACHY measurements for a particular month per year when they fit the model, and then ignore them when they don't. The SCIAMACHY NO2 results as shown in the supplement are almost significant at the 2-sigma level (they are certainly significant at 1-sigma) and are in the opposite direction of what the model shows. The easiest solution would be for the authors to conclude that the SCIAMACHY measurements, when plotted as one month per year, simply aren't up to this, and therefore need to be dropped from this figure entirely. The SCIAMACHY results as shown in Figure 2 and 5 certainly do demonstrate the value of this measurements when they are not subsampled as in Figure 12.

To address the criticism, we added the SCIAMACHY data for $NO_2$ and $O_3$, which showed insignificant gradients, to Fig. 12c,d (as mentioned above). We plotted statistically significant (2-sigma) linear changes as solid lines and insignificant changes as dashed lines.

We also rewrote the explanations related to Fig. 12:

- We mention that SCIAMACHY measurements do not yield statistically significant gradients for the time series of Januaries and Februaries in **P19 L2-3**: 'SCIAMACHY measurements show statistically insignificant changes of $NO_2$ and $O_3$ during Januaries and Februaries (Fig. 12c,d, Supplements Fig. S4)'.

- We also added that contrary to model simulations, SCIAMACHY measurements do not show a $NO_2$ decrease and an $O_3$ increase when analysing changes for any particular calendar month (**P19 L3-5)**: "Contrary to the TOMCAT simulations, SCIAMACHY measurements do not show a statistically significant $NO_2$ decrease and $O_3$ increase when analysing changes for any particular calendar month'.

- We also discuss possible reasons for the model-measurements differences (Fig. 12c,d) on **P19 L5-11**: "From September to February, the gradient of $O_3$ time series increases, becoming more positive for both SCIAMACHY and TOMCAT data, resulting for February in small, statistically insignificant negative gradients for SCIAMACHY observations and small but statistically significant positive gradients for TOMCAT. Similarly for $NO_2$ mixing ratios, from September to February the gradients decrease i.e. they become more positive for both, SCIAMACHY and TOMCAT results. The SCIAMACHY data show larger errors on gradients of the time series for individual months, than those of the TOMCAT model. This results from the stronger oscillating structure in the SCIAMACHY time series. The reasons for the observed oscillations and their strength are not yet unambiguously identified and are under investigation."

Page 19 line 13- This paragraph purports to explain the absence of change in AoA. While it is certainly possible that one could have a change in N2O and not a change in AoA, this point has not been proven. At the same time, the explanation seems to be simply a complicated statement of the fact that changes in N2O are governed by changes in upwelling speed, which obviously couple to AoA. Unless the authors can offer some additional insight here I would recommend dropping this paragraph.

We reworked the explanation of $N_2O$-AoA non-linear relation on **P19 L19-33 as follows**:

'The negative AoA gradients for the 2004-2012 period during the boreal winter months (January and February) and positive AoA gradients during the boreal autumn months (September and October) cancel, i.e. there is no statistically significant linear change/gradient in the annual mean AoA (Fig. 8b). In contrast, the monthly gradients over the same periods for the chemical species $N_2O$, $NO_2$ and, as a result of the $NO_x$ ozone catalytic destruction cycle, $O_3$ do not cancel in the annual means. This effect is primarily attributed to the non-linear relationship between AoA and $N_2O$. This is explained by the following: 1) AoA strongly depends on the speed of the BDC, with lower AoA values indicating an acceleration, and higher AoA indicating deceleration of the vertical transport. In the absence of significant photolytic loss of $N_2O$ via the Reaction (R7), the changes in stratospheric $N_2O$ would be controlled only by changes of the rate of the tropical upwelling of the BDC (or simply by AoA), i.e. faster upwelling would enhance transport of $N_2O$ to the stratosphere, and vice versa. Without photolytic loss, the rate of change of $N_2O$ concentration would be inversely proportional to the AoA change; 2) the dominant chemical loss mechanism of $N_2O$ is through its photolysis. The amount of photolysed $N_2O$ depends on the residence time of $N_2O$ and this in turn depends on the transport speed, i.e. AoA. Longer residence times of $N_2O$ result from a transport slow-down. Consequently, there is more time for photolytical destruction of $N_2O$; 3) as the amount of $N_2O$ is controlled by both transport and photochemistry, its changes do not cancel in the annual average; 4) the amount of $NO_2$ and $O_3$ are chemically linked to that of $N_2O$. Overall, the changes of $NO_2$ and $O_3$ are dependent on both the amount of $N_2O$ transported to the stratosphere and its residence time'.

Supplement – The notation of SCIAMACHY, TOMCAT, and Insignificant is confusing, since the gray Insignificant lines can be either of the former two. The current notation obscures the important fact that the subdivided SCIAMACHY measurements never show a significant trend in the opposite direction to the overall trend in N2O and O3. I recommend using just green, blue, and, if the authors think it is helpful, a dotted version of these colored lines for an insignificant trend.

We reworked Fig. S4-S7, i.e. we plotted statistically significant (2-sigma) changes as solid lines, and insignificant changes as dashed lines.

[Figure]

Figure 12. Linear changes of AoA, $N_2O$, $NO_2$, and $O_3$ minus QBO effect averaged over (a-d) Februaries 2004-2012 and (e-h) Septembers 2004-2011 in the tropical stratosphere between 30 and 35 km altitude. Colour coding indicates the data source: TOMCAT CNTL simulation (green), and SCIAMACHY measurements (dark blue). Colour-coded trend values and their errors (in % per decade) are shown in each panel. Solid lines indicate statistically significant linear changes at the 2σ level, dashed lines indicate statistically insignificant changes.

---

## Author Comment (AC2) · 15 Nov 2018

We thank the Referee for the time spent on reading and reviewing this manuscript, as well as raising some important points. Below we address these points one by one. Our responses are highlighted in blue. We refer to the manuscript using, for instance, **P1 L12**, which means page 1, line 12.

The authors aimed to understand the negative ozone change seen in the middle tropical stratosphere, and in doing so made the link that increases in NO2 as a result of dynamical changes were causing the loss of ozone in the region of focus around 30-35 km. However, they were not able to link this to a statistically significant change in the age of air, which is also an interesting result. Nevertheless, the importance of understanding multiple chemical and dynamical drivers in the stratosphere is highlighted and the authors present interesting results and raise questions worth investigating further.

Although annual changes in AoA are statistically insignificant we discovered that seasonal changes in AoA are significant and result in specific physico-chemical mechanisms that control the $O_3$ amount and its changes in annual means. The $N_2O$, $NO_2$, and $O_3$ responses to the changing BDC and AoA are non-linear. We changed the text to point out this issue more strongly, i.e. on **P19 L19-33.**

However, my concern is that some of the points made, and hypotheses, are not well supported by what is presented, or the authors are not explicit and careful with how they present results (e.g. correlation coefficient, below).

I think this work is useful, and should be published, but changes are needed to make it explicitly clear what (i) can definitely be said from the observations, model and comparisons, (ii) what are the hypotheses the authors are putting forward, and (iii) what are the clear open questions that need to be addressed in future.

To address this comment, we rewrote the discussion of model-satellite comparison in Sect. 3.4. SCIAMACHY data, yielding statistically significant and insignificant gradients are both plotted in Fig. 12 (see our reply to Reviewer #1). The discussion of the possible reasons for the differences between the model and measurements has been rewritten (**P19 L5-11**). We also explained better our hypotheses of the non-linear relationship between AoA and $N_2O/NO_x/O_3$ (**P19 L19-33**). Concerning the issue (iii) mentioned by the Referee "what are the clear open questions that need to be addressed in future" we explicitly described in the last two paragraphs of the Summary (**P20 L20-P21 L5**), i.e. possible causes of the observed seasonal AoA variations.

Comments: 1. I am in agreement with the other referee that the non-significant, even opposite signal (and sign of trend) in February, though non-significant (in the supplement), is not addressed head on. Data is often messy and difficult to deal with especially when

comparing with a model, and should be presented front and centre even if there is a contradiction or lack of evidence to contend with. This actually requires a deeper discussion, because if the model disagrees with the data in the sign of the trend (and it appears consistent between NO2 and O3 in February in the supplementary materials despite the non-significance) then that raises questions that need to be highlighted (for example, is it a model or an observational problem?). I won't labour on this point further, or repeat points raised by the other referee, as the other referee has spent quite some time on points related to this.

We agree with the Referee in his criticism and we replotted Fig.12. We included the SCIAMACHY measurements yielding insignificant gradients in Fig. 12c,d; we noted statistically significant at 2-sigma level changes as solid lines, and insignificant changes as dashed lines (see also our reply to Reviewer #1).

We also rewrote the explanations of the behaviour observed in Fig. 12:

- We mention that SCIAMACHY measurements do not show statistically significant changes for $NO_2$ and $O_3$ time series of Januaries and Februaries in **P19 L2-3**: 'SCIAMACHY measurements show statistically insignificant changes of $NO_2$ and $O_3$ during Januaries and Februaries (Fig. 12c,d, Supplements Fig. S4)'.

- We also mentioned that contrary to model simulations, SCIAMACHY measurements do not show a $NO_2$ decrease and an $O_3$ increase when analysing changes for any particular calendar month (**P19 L3-5**): "Contrary to the TOMCAT simulations, SCIAMACHY measurements do not show a statistically significant $NO_2$ decrease and $O_3$ increase when analysing changes for any particular calendar month'.

- We also discuss possible reasons for the model-measurements differences (Fig. 12c,d) on **P19 L5-11**: 'From September to February, the gradient of $O_3$ time series increases, becoming more positive for both SCIAMACHY and TOMCAT data, resulting for February in small, statistically insignificant negative gradients for SCIAMACHY observations and small but statistically significant positive gradients for TOMCAT. Similarly for $NO_2$ mixing ratios, from September to February the gradients decrease i.e. they become more positive for both, SCIAMACHY and TOMCAT results. The SCIAMACHY data show larger errors on gradients of the time series for individual months, than those of the TOMCAT model. This results from the stronger oscillating structure in the SCIAMACHY time series. The reasons for the observed oscillations and their strength are not yet unambiguously identified and are under investigation'.

2. Page 4, L20-23: is this relationship specifically in the 30-35 km tropical region of the study (see comment 2 below).

The sentence on **P4 L20-23** could indeed be misleading the way it is. We removed the reference of Plummer et al. (2010) because he was dealing with tropical, but lower stratosphere. However, we leave the reference of Kracher et al. (2016), in the manuscript as we consider that this research addresses the impact of tropical upwelling on the $N_2O$ lifetime. Also, to avoid the confusion with regard to their results, we rewrote **P4 L20-23** as follows: 'While accelerated tropical upwelling enhances transport of $N_2O$ from its source towards the stratosphere, it reduces its lifetime (e.g. Kracher et al., 2016). The amount of $NO_x$ is then affected by a shorter $N_2O$ residence time causing its lower production via Reaction (R8a), and as a consequence less $O_3$ loss in the tropical mid-stratosphere'.

3. Page 4, L26: actually I would argue that the decrease Kyrola et al., 2013 found was up to 6-8% at its core (Fig. 16), which is more in line with that quoted for Gebhardt et al 2014. However, the core of the negative region in Gebhardt et al., 2014 is upward of -18% (Fig. 8).

Fig. 16 of Kyrölä et al. (2013) shows the change of $O_3$ trends between the two periods. In our manuscript, we refer to $O_3$ change during the specific period 1997-2011 from Kyrölä et al., 2013, as it is the closest to our period 2004-2012. Consequently, we believe Fig. 15 from Kyrölä et al. (2013) is the most suitable. We improved the sentence on **P4 L26-28** as follows: 'Kyrölä et al. (2013, Fig.15) showed a statistically significant negative trend of $O_3$ of around 2-4% per decade in the tropical region (10° S-10° N) at altitudes 30-35 km for the period 1997-2011 from the combined Stratospheric Aerosol and Gas Experiment (SAGE) II-Global Ozone Monitoring by Occultation of Stars (GOMOS) dataset '.

Could the authors be clear in what they mean here since I believe the -10% refers to the 20S-20N (Fig 7) profile; since the authors focus in on +/-10 deg. latitude region, the higher value seems more appropriate but then the estimate in this manuscript is almost 2x smaller.

We mixed up the 10°S-10°N defined as the tropical region in our study with the 20°S-20°N region used in other studies, e.g. Gebhardt et al. (2014). Since we provided the definition of tropics on **P3 L5-6** as 10°S-10°N, we modified the sentence on **P4 L26-31** as follows: 'Kyrölä et al. (2013, Fig.15) showed a statistically significant negative trend of $O_3$ of around 2-4% per decade in the tropical region (10° S-10° N) at altitudes 30-35 km for the period 1997-2011 from the combined Stratospheric Aerosol and Gas Experiment (SAGE) II-Global Ozone Monitoring by Occultation of Stars (GOMOS) dataset. Gebhardt et al. (2014, Fig.8) identified much stronger negative $O_3$ trend of up to 18% per decade in the same altitude and latitude range for the period August 2002-April 2012 from SCanning Imaging Absorption spectroMeter for Atmospheric CHartographY (SCIAMACHY) observations'.

I assume, though perhaps the authors should check, this difference is due to a different time period and set of regressors used? At the very least please be explicit about what region the numbers represent and are comparable to the region focused on in this manuscript.

To address this point, we now say on **P8 L22** that SCIAMACHY $O_3$ changes were 'reaching 12% per decade' rather than 'reaching around 10% per decade'. We would also like to highlight that Gebhardt et al. (2014) applied SCIAMACHY limb $O_3$ scientific dataset v2.9, which was suffering from a drift. In our research we use the $O_3$ scientific dataset v3.5 (as mentioned on **P6 L21**), which is drift-corrected in contrast to v2.9.

4. Page 8, L4-9: I am not sure I agree that it is consistent to ignore the monthly autocorrelation when using all months. It seems to me consistent not to use it for single month (i.e. Jan only, etc) estimates (since there should be no autocorrelation between the months 12 months apart) and to indeed consider autocorrelation for the full time series since that is typically the case if they are next to each other in a continuous time series. This is only reasonable if you can state explicitly that there is no change in the significance - does considering it have an effect on your conclusions?

Autocorrelation of the noise affects errors of the trends but does not affect the value of the trends themselves. As the major focus of current manuscript is the seasonal changes of transport and chemical compounds, the use of autocorrelation of the noise is not needed. We do not apply it in our Multivariate Linear Regression applied to the annual averages. In Fig. 2 and Fig. 3a,d our focus is on the similarities of the observed patterns of the SCIAMACHY measurements and TOMCAT model in the tropical mid-stratosphere. Nedoluha et al. (2015) , who analysed tropical $O_3$ trends from HALOE and MLS, also did not apply an autocorrelation term.

5. Page 9, L14: the inference the authors make from Fig.3 is that chemistry has little impact on the 30-35 km tropical region; for O3 and N2O I think this is reasonable. But for 3d-f it seems that in the box, NO2 is roughly split 70/30 or maybe 50/50 in the peak positive change. So it isn't clear to me if this statement is fully backed up by the plot (or perhaps its a non-linear interaction?). Please could the authors comment on this, perhaps with values.

For the simulations used in the fixed dynamical (fDYN) case, $N_2O$ (Fig. 3i) shows statistically significant but weak positive changes in the tropical mid-stratosphere. Consequently, an increase of $NO_2$ (Fig. 3f) is also expected due to Reaction (R8a), $N_2O +O(^1D)$. As a result, a small statistically significant $NO_2$ increase in the tropical mid-stratosphere (~3% per decade), caused by the chemical mechanism, does lead to a statistically significant $O_3$ decrease. However in the fSG TOMCAT simulation, $NO_2$ shows positive changes in the tropical mid-stratosphere (Fig. 3e) similar to TOMCAT CNTL simulation (Fig. 3d) and SCIAMACHY measurements (Fig. 2b). We infer that the major impact of the positive changes of $NO_2$ comes from the dynamics i.e. the slower transport of $N_2O$. We provided minor correction on **P10 L1** from '...around 1-3 % per decade' to '...around 3 % per decade'.

6. While Fig 4a. shows a combined non-linear shape, it appears that the anticorrelation (linear slope for each level) reduces with higher altitude, being almost flat at 35 km (green). Why does this happen? Does this indicate that the mechanism proposed is no longer operating as efficiently in the upper part of the box?

We indeed found the drop of anti-correlation between $N_2O$ and $NO_2$ at the altitude of around 35 km. Although, $N_2O$ and $NO_2$ on average highly anti-correlate in the tropical middle stratosphere (r=-0.9, Fig. 6). This anti-correlation becomes weaker at 35 km altitude in the tropics during May-July and November and anti-correlation varies from -0.52 to -0.57 (these results are not included in the manuscript). In particular, at altitudes above 35 km, produced NO (via Reaction R8a) reacts rapidly with N (NO + N -> $N_2$ +O) and therefore converts NO back to N2. Therefore the $N_2O$-$NO_2$ anti-correlation becomes weaker in the upper edge of our target altitude region and above.

7. Fig 6, 10, and all discussion related to the R^2 statistic: this is very confusing and needs to be stated explicitly and correctly. R^2 is formally the "coefficient of determination", which can be the square of, but not same as the "correlation coefficient". Further R^2 can only range from 0 to 1, while the correlation coefficient can range from -1 to +1. Please check all instances of this and be correct in its usage; in many places this is confusing and leads the reader to have to try and work out what the authors mean.

We agree with the Referee that $R^2$ was misleading and we removed $R^2$ from the text of manuscript entirely and reformulated the sentence on **P12 L15-16** as follows: 'Recognising the tight relationships within the tropical mid-stratosphere $N_2O$-$NO_x$-$O_3$ chemistry, seen in Figs. 4 and 5, we further calculated Pearson correlation coefficients, between the chemical species as well as with the dynamical AoA tracer'.

8. Page 16, L5: 0.6 is an arbitrary threshold; please state this explicitly.

We improved the sentence as suggested on **P16 L4-5**: 'Horizontal dashed lines indicate an arbitrary threshold of moderate correlation, which is represented by the value of -0.6.'
We also corrected the caption of Fig. 10 accordingly.

9. Fig. 11: is this also integrated over 10S-10N?

Yes, to avoid any misunderstanding we rephrased the caption of Fig. 11 as follows:
'Annual cycle of monthly mean $N_2O$ (ppbV, contours, 15 ppbV interval) and AoA (years, colours, 0.2 yr interval) as a function of altitude from TOMCAT run CNTL in the tropical region, averaged over the period January 2004–April 2012.'

10. Page 19, L4-5: Is this a hypothesis or a demonstrable fact? I do not understand why it is a limitation of the measurements, given the description earlier of the limb observations being well-distributed in the tropics and the period being considered is the same for the model data. If the effect is demonstrable, then this would provide good evidence the model is correct and why we don't have to worry about the insignificance and/or inverse correlations. If it is a hypothesis, please state explicitly this is the case.

We reworked the hypothesis of larger errors of SCIAMACHY gradients/linear changes on **P19 L8-11** as follows: 'The SCIAMACHY data show larger errors on gradients of the time series for individual months, than those of the TOMCAT model. This results from the stronger oscillating structure in the SCIAMACHY time series. The reasons for the observed oscillations and their strength are not yet unambiguously identified and are under investigation'.

11. Page 19, L16-22. I'm afraid I found this explanation difficult to follow. Please rewrite to be clearer. Is the summary that the N2O "changes do not cancel in the yearly average" because photolysis has an affect that AoA is not impacted by?

We reworked the explanation of $N_2O$-AoA non-linear relation on **P19 L19-33** as follows (see also reply to reviewer #1): 'The negative AoA gradients for the 2004-2012 period during the boreal winter months (January and February) and positive AoA gradients during the boreal autumn months (September and October) cancel, i.e. there is no statistically significant linear

change/gradient in the annual mean AoA (Fig. 8b). In contrast, the monthly gradients over the same periods for the chemical species $N_2O$, $NO_2$ and, as a result of the $NO_x$ ozone catalytic destruction cycle, $O_3$ do not cancel in the annual means. This effect is primarily attributed to the non-linear relationship between AoA and $N_2O$. This is explained by the following: 1) AoA strongly depends on the speed of the BDC, with lower AoA values indicating an acceleration, and higher AoA indicating deceleration of the vertical transport. In the absence of significant photolytic loss of $N_2O$ via the Reaction (R7), the changes in stratospheric $N_2O$ would be controlled only by changes of the rate of the tropical upwelling of the BDC (or simply by AoA), i.e. faster upwelling would enhance transport of $N_2O$ to the stratosphere, and vice versa. Without photolytic loss, the rate of change of $N_2O$ concentration would be inversely proportional to the AoA change; 2) the dominant chemical loss mechanism of $N_2O$ is through its photolysis. The amount of photolysed $N_2O$ depends on the residence time of $N_2O$ and this in turn depends on the transport speed, i.e. AoA. Longer residence times of $N_2O$ result from a transport slow-down. Consequently, there is more time for photolytical destruction of $N_2O$; 3) as the amount of $N_2O$ is controlled by both transport and photochemistry, its changes do not cancel in the annual average; 4) the amount of $NO_2$ and $O_3$ are chemically linked to that of $N_2O$. Overall, the changes of $NO_2$ and $O_3$ are dependent on both the amount of $N_2O$ transported to the stratosphere and its residence time'.

[Figure]

Figure 12. Linear changes of AoA, $N_2O$, $NO_2$, and $O_3$ minus QBO effect averaged over (a-d) Februaries 2004-2012 and (e-h) Septembers 2004-2011 in the tropical stratosphere between 30 and 35 km altitude. Colour coding indicates the data source: TOMCAT CNTL simulation (green), and SCIAMACHY measurements (dark blue). Colour-coded trend values and their errors (in % per decade) are shown in each panel. Solid lines indicate statistically significant linear changes at the 2σ level, dashed lines indicate statistically insignificant changes.

---

## Author Comment (AC3) · 15 Nov 2018

I do not have much to add to the reviewers comments, except the following:

Their discussion on lines 7-11 on page 5 reads as if they are contradicting themselves. Thus line 7 says "decrease in N2O" while lines 8-10 discuss an increase in upwelling leading to "lower N2O oxidation" which necessarily would produce an increase in N2O. It is true that the specific model perturbation we introduced (Nedoluha et al., 2015b) had an increase in upwelling; however, the model-to-model comparison we made was to show that upwelling strength varies directly as N2O and and inversely as NOy. And the objective was to explain the lower ozone, which would result from weaker upwelling. I would therefore like to suggest a wording change to be clearer:

Using a 2D chemical-dynamical model, they showed that changes to the tropical upwelling could lead to changes in the N2O oxidation via (R8a) and thus affect the NOy production. Based on this, Nedoluha et al. (2015b) concluded that weaker tropical upwelling could therefore explain the decrease of O3 in the tropical mid-stratosphere.
David Siskind

We thank David Siskind for his helpful comment. We have improved the text as suggested on **P5 L9-11** as follows: 'Using a 2D chemical-dynamical model they showed that the changes in the tropical upwelling could lead to the changes in the $N_2O$ oxidation via (R8a) and thus affect $NO_y$ production. Based on this, Nedoluha et al. (2015b) concluded that weaker tropical upwelling could, therefore, explain the decrease of $O_3$ in the tropical mid-stratosphere'.

---

## Author Response (AR2)

We thank the Referee #1 for the comments on our manuscript. We describe our responses and improvements below in blue.

The manuscript has been improved and the seasonal differences in the trends are a very interesting result. I have only two points which the authors should address.

Most importantly, the rewritten explanation for the "non-linear relationship between AoA and N2O" (page 19 lines 19-33) seems to me to be no clearer than the previous version (I note that Reviewer 2 also seems confused by this portion of the manuscript). The authors seem to be suggesting that photolytic loss (R7) somehow has a different relationship to AoA than the reaction of N2O with O(1D) (R8a and b), but it is still not clear to me from the explanation why these are different. Both reactions will result in greater N2O destruction with longer residence time (i.e. slower transport). The authors state that "Without photolytic loss, the rate of change of N2O concentration would be inversely proportional to the AoA change". Why isn't this the case for photolytic loss as well? Unless the authors can clearly express why R7 and R8 are differently affected by AoA this explanation is simply not useful and should not be included in the manuscript.

We acknowledge that our revised explanation was still suboptimal and clearly needs to be improved as the reviewer was not able to understand it. In fact, we did not attempt to show 'that photolytic loss (R7) somehow has a different relationship to AoA than the reaction of N2O with O(1D) (R8a and b)'. Thus contrary to the impression we gave, we recognise that photolytic loss of $N_2O$ and reaction of $N_2O$ with $O(^1D)$ have the same relationship to AoA. The Referee is absolutely right that both photolysis and reaction with $O(^1D)$ 'will result in greater N2O destruction with longer residence time (i.e. slower transport)'. We would like to highlight here, that photolysis destroys ~90% of all stratospheric $N_2O$ (**P4 L5-7** and **Fig. 7b**) in the region of study and does not produce $NO_x$ species. The remaining ~10% loss of $N_2O$ is via the reaction with $O(^1D)$, which in most cases leads to NO production (**P4 L9-10** and **Fig. 7b**).

Theoretically, for a case when photochemical loss of $N_2O$ is absent, the change of $N_2O$ would be roughly inversely proportional to the AoA change. Then with increase of AoA (or other words transport slowdown) there would be less $N_2O$ transported from the lower altitudes. However, for a case with photochemical loss, the amount of $N_2O$ depends not only on its transport from the troposphere, but also on its residence time at a given altitude. If the transport slows down, then the loss of $N_2O$ would be more pronounced in comparison to the idealised 'without photochemical loss' case, as it would be defined not only by transport speed, but also photochemistry via Reactions (R7) and (R8a,b).

We address changes on **P19 L25-35** as follows:

'The negative AoA gradients for the 2004-2012 period during the boreal winter months (January and February) and positive AoA gradients during the boreal autumn months (September and October) cancel, i.e. there is no statistically significant linear change/gradient in the annual mean AoA (Fig. 8b). In contrast, the monthly gradients over the same periods for the chemical species $N_2O$, $NO_2$ and,

as a result of the $NO_x$ ozone catalytic destruction cycle, $O_3$ do not cancel in the annual means. This effect is primarily attributed to the non-linear relationship between AoA and $N_2O$. That implies that for example the slowdown in tropical upwelling would indicate (1) slower $N_2O$ transport from lower altitudes, which would lead to a decrease of $N_2O$, and (2) destruction of more $N_2O$ by photolysis (R7, around 90%) and $O^1D$ (R8a and R8b, around 10%), as the residence time of $N_2O$ in this region becomes longer. In contrast, the speedup in tropical upwelling would mean that more $N_2O$ is transported from the lower altitudes, and its residence time is shorter. This leads to lower $N_2O$ destruction via photochemistry. The amounts of $NO_2$ and $O_3$ are chemically linked to that of $N_2O$. Overall, the changes of $NO_2$ and $O_3$ are dependent on both the amount of $N_2O$ transported to the stratosphere and its residence time.'

Page 1 line 16 – "Although the changes in AoA cancel out when averaging over the year …". As far as I can tell the authors have changed "no trend" (expressed here as "cancel out") to "no statistically significant trend" everywhere in the paper, except here. Please replace "cancel out" with "no statistically significant trend".

[revised manuscript text omitted]